# LoRA Dropout as a Sparsity Regularizer for Overfitting Reduction

## Abstract

Parameter-efficient fine-tuning methods, represented by LoRA, play an essential role in adapting large-scale pre-trained models to downstream tasks. However, fine-tuning LoRA-series models also faces the risk of overfitting on small training datasets, and there's still a lack of theoretical guidance and practical mechanisms to control overfitting on LoRA-based PEFT methods. This paper introduces a novel dropout-based sparsity regularizer for LoRA, dubbed LoRA Dropout, which mitigates overfitting by applying refined dropout to LoRA's low-rank matrices. We establish a theoretical framework that models dropout in LoRA as a sparse fine-tuning process and derive a generalization error bound under this sparsity regularization. Theoretical results show that appropriate sparsity can tighten the gap between empirical and generalization risks and thereby control overfitting. We further enhance the sparsity patterns in conventional dropout methods and propose an innovative LoRA Dropout method for more precise sparsity regularization to achieve better overfitting reduction. Furthermore, we introduce a test-time ensemble strategy and provide theoretical evidence demonstrating that the ensemble method can further compress the error bound and lead to better performance. Extensive experiments on various NLP tasks validate the effectiveness of our LoRA Dropout framework in improving the model's performance.

## 1 Introduction

In recent years, Pre-trained Language Models (PLMs) (Devlin et al., 2018; Liu et al., 2019; He et al., 2020; Touvron et al., 2023) have demonstrated increasingly superior performances in various NLP tasks as the rapid growth of model parameter scale. However, with the increasing model capacity and complexity, the challenge arises when adapting the PLMs to specific downstream tasks, as fully fine-tuning often requires substantial computational resources. Therefore, a new fine-tuning paradigm emerges named Parameter-Efficient Fine-Tuning (PEFT), aiming to adapt PLMs to specific downstream tasks with minimal adjustments to their parameters.

Among the works in the field of PEFT (Houlsby et al., 2019; Lester et al., 2021; Li & Liang, 2021; Hu et al., 2021; Zhang et al., 2023; Ma et al., 2024), the Low-Rank Adaptation (LoRA) method (Hu et al., 2021) and its variants (Dettmers et al., 2023; Zhang et al., 2023; Zi et al., 2023) have been the most effective and widely adopted. The basic idea behind LoRA is that only some zero-initialized delta weight matrices get optimized during fine-tuning, and the original pre-trained parameters remain unmodified. To improve parameter efficiency, LoRA further decomposes the delta weight matrix into the product of two low-rank matrices.

However, one significant challenge when fine-tuning LoRA-series models on downstream tasks is **overfitting**. As shown in Figure.1(a), the gap between train and test losses on both LoRA becomes larger during fine-tuning, indicating a strong overfitting tendency of LoRA training. Simply reducing the rank of LoRA (i.e. reducing learnable parameters) could help alleviate overfitting, but fewer learnable parameters indicate less expressive power, and might lead to suboptimal performances. Therefore, it is hard to select a proper rank that could balance the expressiveness and overfitting risk. AdaLoRA (Zhang et al., 2023) proposes to automatically prune unimportant parameters with learned importance scores during training to prevent overfitting. However, this parameter selection method heavily relies on gradients of parameters on the training data, which in turn makes the selected

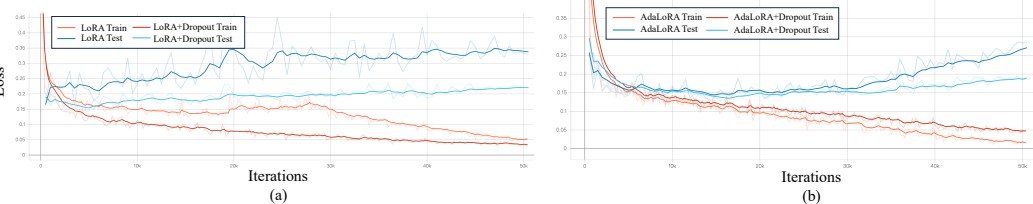

Figure 1: Loss curves on train and test set of SST2 dataset during fine-tuning of (a) LoRA w/wo our dropout framework, (b) AdaLoRA w/wo our dropout framework.

parameters less generalizable to unseen test data and increases the risk of overfitting. This is also practically demonstrated by the loss curves in Figure.1(b).

The dropout regularization (Hinton et al., 2012; Srivastava et al., 2014) is known as one of the most popular and effective techniques in deep learning to control overfitting. It works by randomly masking neurons and does not require to reduce parameter budget. Recently there have been works attempting to combine dropout with LoRA to control the overfitting risk when fine-tuning (Wang et al., 2024). However, these works fail to answer the following profound question:

*What is the theoretical mechanism behind the alleviation of overfitting on the training data through random dropout of LoRA parameters?*

In this paper, we answer the above question and build our theoretical framework by modeling the training process with dropout from the perspective of sparse fine-tuning, and show that fine-tuning LoRA with dropout can be viewed as an optimization problem with sparsity regularization. We further provide a generalization error bound under the sparsity regularization framework. Through this bound, we reveal that introducing appropriate sparsity on LoRA tunable parameters during fine-tuning helps to balance the empirical risk minimization and complexity of the adaptation function. Therefore, proper dropout would help tighten the gap between empirical and generalization risks and reduce overfitting on the training data.

Furthermore, with our theoretical framework, we identify the flaws in the original dropout method and improve upon it. The theoretical analyses point out a crucial condition for effectively applying the dropout mechanism in LoRA training, namely the sparsity condition. While most prior studies have implemented dropout on hidden representations or at the token level (Wang et al., 2024), they fall short in generating a sufficiently diverse sparsity pattern across the parameter space. This limitation hampers the precision of sparsity regularization. To address this issue, we introduce an innovative approach called LoRA Dropout, which enhances the dropout process by incorporating more expressive sparsity patterns for the updated parameters—without significantly increasing GPU memory overhead. We also propose a test-time ensemble method with an ensemble classifier consisting of models with different parameter dropouts. This ensemble method can further improve the model's test-time performance, which is also supported by theoretical evidence.

In summary, we conclude the main contributions of this paper as follows. We provide theoretical analyses on the generalization error bound of dropout on LoRA, balancing the empirical risk minimization and complexity of the adaptation function class through sparsity regularization. Based on the theoretical evidences, we propose a novel dropout framework for LoRA-based models. With expressive sparsity patterns, our LoRA Dropout method can effectively promote model's generalization ability on downstream tasks. Moreover, we propose an ensemble strategy during the inference stage, and show that this strategy will lead to an ensemble model with a tighter error bound and further enhance the model's test-time generalizability. Extensive experiments conducted on a wide range of NLP tasks demonstrate the effectiveness of our method in improving the model's performances.

## 2 HOW DOES DROPOUT BALANCE OVER- AND UNDER-FITTING IN LORA

In this section, we present the theoretical analyses of the LoRA optimization process with dropout to reveal how dropout alleviates overfitting in LoRA fine-tuning. We first model the LoRA fine-tuning with dropout as an optimization problem under model sparsity regularization. Then we propose the

generalization error bound under the sparsity regularization framework and reveal the theoretical mechanism behind the trade-off between underfitting and overfitting of fine-tuning with dropout.

## 2.1 FINE-TUNING WITH DROPOUT THROUGH THE LENS OF SPARSE REGULARIZATION

Supposing a pre-trained model $\mathcal{M}^0$ parameterized by $\boldsymbol{\theta}^0 \in \mathbb{R}^d$, LoRA-like methods tune the model $\mathcal{M}^0$ with low-rank parameterization of the delta parameters $\Delta\boldsymbol{\theta}$. Dropout (Srivastava et al., 2014) is a widely adopted mechanism in traditional deep neural network training for overfitting risk reduction. The core idea of dropout can be formulated as a **diversified sparse activation of neurons** (Gal & Ghahramani, 2016). Hence, inspired by (Fu et al., 2023), we model the LoRA fine-tuning with random dropout as an optimization problem under sparsity regularization on parameter space. With dropout that samples random neurons of LoRA matrices with a probability $p$ and masks them to zeros, the updated $\Delta\boldsymbol{\theta}$ enjoys a sparsity property that each entry in the product of the LoRA matrices will be zero with probability $p$. Let us denote $\boldsymbol{\theta} = \boldsymbol{\theta}^0 + \Delta\boldsymbol{\theta}$ as the fine-tuned model parameters, where $\Delta\boldsymbol{\theta}$ is realized by LoRA reparameterization with dropout. Assume $\boldsymbol{d} \in \{0,1\}^d$ as a dropout instance applied to the production of LoRA matrices (i.e., $\Delta\boldsymbol{\theta}$) sampled from a Bernoulli distribution, i.e., $\boldsymbol{d} \sim \text{Bern}(p)$, where 1 denotes the corresponding entry is dropped to zero. The fine-tuning objective can be formulated as:

$$\min_{\Delta\boldsymbol{\theta}} \mathcal{L}(\boldsymbol{\theta}^0 + \Delta\boldsymbol{\theta}), \text{ s.t. } \mathbb{E}_{\boldsymbol{d}\sim\text{Bern}(p)}||\boldsymbol{d} \odot \Delta\boldsymbol{\theta}||_2^2 \le c, \tag{1}$$

where $c$ is a constant, and the condition denotes the sparsity of $\Delta\boldsymbol{\theta}$. By Lagrange duality, problem (1) can be rewritten as:

$$\hat{\mathcal{L}} = \min_{\Delta\boldsymbol{\theta}} \max_{\lambda} \mathcal{L}(\boldsymbol{\theta}^0 + \Delta\boldsymbol{\theta}) + \lambda\mathbb{E}_{\boldsymbol{d}\sim\text{Bern}(p)}||\boldsymbol{d} \odot \Delta\boldsymbol{\theta}||_2^2, \tag{2}$$

and $c$ is eliminated as a constant. Hence, we formulate the regularized optimization problem:

$$\min_{\Delta\boldsymbol{\theta}}\mathcal{L}_\lambda = \min_{\Delta\boldsymbol{\theta}} \mathcal{L}(\boldsymbol{\theta}^0 + \Delta\boldsymbol{\theta}) + \lambda\mathbb{E}_{\boldsymbol{d}\sim\text{Bern}(p)}||\boldsymbol{d} \odot \Delta\boldsymbol{\theta}||_2^2 \le \hat{\mathcal{L}}. \tag{3}$$

where $\lambda$ is an arbitrary hyperparameter. This optimization objective is upper bounded by $\hat{\mathcal{L}}$, which is equivalent to the optima of problem (1).

## 2.2 GENERALIZATION ANALYSIS

In this subsection, we introduce the stability analysis of a sparse-regularized algorithm to analyze the generalization error bound of dropout fine-tuning through optimizing Eq. (3). Stability has been a widely studied topic in machine learning (Bousquet & Elisseeff, 2002; Charles & Papailiopoulos, 2018; Kuzborskij & Lampert, 2018) and demonstrated as an important property for analyzing the generalization error bound of a random algorithm (Bousquet & Elisseeff, 2002; Elisseeff et al., 2005). Here we adopt one of the commonly used analytic mechanisms, the Pointwise Hypothesis Stability (PHS), which analyzes the perturbation of the optimal model after removing one of the training samples. Following (Charles & Papailiopoulos, 2018), we denote the entire training dataset as $\mathbf{S} = \{x_i\}_{i=1}^n$ and the dataset after removing a sample $x_i$ as $\mathbf{S}^i = \mathbf{S} - \{x_i\}$. We assume that $i \sim \text{U}(n)$ that the removal is sampled from a uniform distribution. We also denote $\boldsymbol{\theta}_\ell(\mathbf{S})$ as the optimal model parameters w.r.t. loss function $\ell$ and dataset $\mathbf{S}$.

**Definition 2.1** (Pointwise Hypothesis Stability (Bousquet & Elisseeff, 2002)). We say that a learning algorithm $\mathcal{M}$ parameterized by $\boldsymbol{\theta}$ w.r.t. a loss function $\ell$ has pointwise hypothesis stability $\beta$, if:

$$\mathbb{E}_{\mathbf{S},i\sim\text{U}(n)} \left|\ell\left(x_i; \boldsymbol{\theta}_\ell(\mathbf{S}^i)\right) - \ell\left(x_i; \boldsymbol{\theta}_\ell(\mathbf{S})\right)\right| \le \beta, \tag{4}$$

where $\ell(x_i; \boldsymbol{\theta})$ denotes the sample loss of $x_i$ when the model parameter is $\boldsymbol{\theta}$. Here we present a PHS upper bound of LoRA fine-tuning with dropout.

**Proposition 2.2** (PHS Upper Bound of LoRA Fine-tuning with Dropout). *If the loss function $\mathcal{L}_\lambda$ of the algorithm $M$ is $\eta$-Lipschitz, and $\boldsymbol{\theta}_{\mathcal{L}_\lambda}(\mathbf{S}^i)$ is close to $\boldsymbol{\theta}_{\mathcal{L}_\lambda}(\mathbf{S})$, $\boldsymbol{\theta}_{\mathcal{L}_\lambda}(\mathbf{S})$ is close to $\boldsymbol{\theta}_{\mathcal{L}_\lambda}(\mathbf{S}^i)$ whose gap is bounded by a small constant $\epsilon \to 0$, i.e., $||\boldsymbol{\theta}_{\mathcal{L}_\lambda}(\mathbf{S}) - \boldsymbol{\theta}_{\mathcal{L}_\lambda}(\mathbf{S}^i)|| \le \epsilon \to 0$, and the regularization coefficient $\lambda \ge \max\{0, \frac{\eta}{np\epsilon} - \frac{1}{2p}\Lambda_{\min}\}$, where $\Lambda_{\min}$ is the minimum eigenvalue of the Hessian $\nabla^2\mathcal{L}(\boldsymbol{\theta}_{\mathcal{L}_\lambda}(\mathbf{S}))$ at $\boldsymbol{\theta}_{\mathcal{L}_\lambda}(\mathbf{S})$, unitary diagonalized as $U\text{diag}(\Lambda)U^{-1}, \Lambda = \{\Lambda_1, \cdots, \Lambda_d\}$ and*

$\Lambda_{\min} = \min\{\Lambda_1, \cdots, \Lambda_d\}$, *then the algorithm optimizing $\mathcal{L}_\lambda$ on $\mathbf{S}$ has an upper bound of pointwise hypothesis stability of:*

$$\mathbb{E}_{\mathbf{S}, i \sim \mathrm{U}(n)} \left| \mathcal{L}_\lambda\left(x_i; \boldsymbol{\theta}_{\mathcal{L}_\lambda}(\mathbf{S}^i)\right) - \mathcal{L}_\lambda\left(x_i; \boldsymbol{\theta}_{\mathcal{L}_\lambda}(\mathbf{S})\right)\right| \leq \frac{2\eta^2}{(\Lambda_{\min} + 2\lambda p)\, n}. \tag{5}$$

*Proof Sketch.* We first analyze the PHS upper bound of an arbitrary optimization algorithm equipped with $\ell_2$-regularizer in Lemma A.1. Then we formulate our training objective with dropout as a similar problem with weighted $\ell_2$-regularizer and take it into Lemma A.1 to finish the proof. See Appendix A.1 for detailed proofs. □

Moreover, existing works (Bousquet & Elisseeff, 2002; Elisseeff et al., 2005) have connected the stability and generalization error bound with the following lemma adopted from (Bousquet & Elisseeff, 2002, Theorem 11).

**Lemma 2.3.** *For any learning algorithm $\mathcal{M}$ having parameter $\boldsymbol{\theta}$ and bounded loss function $\ell$ satisfying $0 \leq |\ell(x) - \ell(x')| \leq C, \forall x, x'$. If $\mathcal{M}$ has a pointwise hypothesis stability $\beta$, with probability $1 - \delta$, we have:*

$$R(\mathcal{M}, \mathbf{S}) \leq \hat{R}(\mathcal{M}, \mathbf{S}) + \sqrt{(C^2 + 12Cn\beta)/(2n\delta)}, \tag{6}$$

where $R(\mathcal{M}, \mathbf{S}) = \mathbb{E}_x \ell(x; \boldsymbol{\theta})$ and $\hat{R}(\mathcal{M}, \mathbf{S}) = \frac{1}{n}\sum_{i=1}^n \ell(x_i; \boldsymbol{\theta})$ denote the empirical risk and generalization risk of algorithm $\mathcal{M}$ running on dataset $\mathbf{S}$, respectively. This indicates that better algorithm stability will reduce the complexity of the adaptation function class. Therefore, invoking the PHS upper bound $\beta$ of the algorithm with dropout in Proposition 2.2 to Lemma 2.3, we have the following theorem that depicts the generalization error bound of LoRA fine-tuning with dropout:

**Theorem 2.4** (Generalization Error Bound of Fine-tuning with Dropout). *Given a dropout rate $p$ and strength of sparsity regularization $\lambda$, if the algorithm $M$ have a $\eta$-Lipschitz loss function $\mathcal{L}_\lambda$, $\boldsymbol{\theta}_{\mathcal{L}_\lambda}(\mathbf{S})$ is $\epsilon$-close to $\boldsymbol{\theta}_{\mathcal{L}_\lambda}(\mathbf{S}^i)$, i.e., $\|\boldsymbol{\theta}_{\mathcal{L}_\lambda}(\mathbf{S}) - \boldsymbol{\theta}_{\mathcal{L}_\lambda}(\mathbf{S}^i)\| \leq \epsilon \to 0$, and the regularization coefficient $\lambda \geq \max\{0, \frac{\eta}{np\epsilon} - \frac{1}{2p}\Lambda_{\min}\}$, where $\Lambda_{\min}$ is the minimum eigenvalue of the Hessian $\nabla^2 \mathcal{L}(\boldsymbol{\theta}_{\mathcal{L}_\lambda}(\mathbf{S}))$ at $\boldsymbol{\theta}_{\mathcal{L}_\lambda}(\mathbf{S})$, unitary diagonalized as $U \operatorname{diag}(\Lambda) U^{-1}$, $\Lambda = \{\Lambda_1, \cdots, \Lambda_d\}$ and $\Lambda_{\min} = \min\{\Lambda_1, \cdots, \Lambda_d\}$, then for some constant $C$, we have with probability $1 - \delta$,*

$$R(M, \mathbf{S}) \leq \hat{R}(M, \mathbf{S}) + \sqrt{\frac{C^2 + \frac{24C\eta^2}{\Lambda_{\min} + 2\lambda p}}{2n\delta}}. \tag{7}$$

This theorem reveals the theoretical mechanism of the trade-off between underfitting and overfitting of dropout. The theorem shows that the complexity of adaptation function class (i.e., the gap between empirical and generalization risks) gets larger as the dropout rate gets smaller. Concretely, when applying LoRA without dropout, the gap will be the largest, which depicts the high risk of overfitting with LoRA fine-tuning. However, when the dropout rate gets larger and tends to 1, it is equivalent to conducting no fine-tuning, which increases the empirical risk and makes the model underfit the training data. Hence, an appropriate dropout mechanism can theoretically balance a trade-off between the empirical risk minimization and the complexity of adaptation function classes, thereby enhancing the test-time performances as well as learning sufficiently from data.

## 3 PROPOSED LORA DROPOUT FRAMEWORK

In spite of our promising theoretical results, there still exists a practical gap between the current dropout fashion and our theoretical setting. Our theoretical framework identifies a significant constraint for effectively applying the dropout mechanism in LoRA tuning, which is the sparsity constraint. However, most prior studies fail to generate a sufficiently diverse sparsity pattern across the parameter space, limiting their practical performance. In this section, we aim to bridge the gap and present our LoRA Dropout framework, which enhances the dropout process by incorporating more expressive sparsity patterns without significantly increasing GPU memory overhead. We start by briefly reviewing the LoRA method. Then we introduce the key insights and details of the LoRA Dropout method. The overall training and testing procedure is summarized in Alg 1.

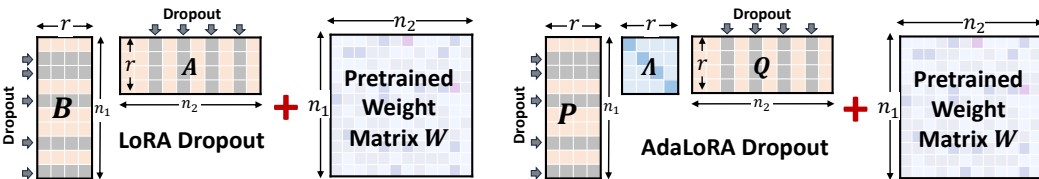

Figure 2: Our proposed LoRA Dropout framework combined with both LoRA and AdaLoRA methods.

## 3.1 BACKGROUND: LOW RANK ADAPTATION (LORA)

Before introducing our method, we give a brief review of the LoRA method (Hu et al., 2021). When fine-tuning on downstream task, to maintain the knowledge from the pre-training period, LoRA freezes the pre-trained parameters $W_0 \in \mathbb{R}^{n_1 \times n_2}$, and updates a zero-initialized delta weight matrix $\Delta W$. To control the number of tunable parameters, as shown in Eq.8, the delta weight matrix $\Delta W$ can be further decomposed into the product of two low-rank matrices, $A \in \mathbb{R}^{r \times n_2}$ and $B \in \mathbb{R}^{n_1 \times r}$, where $r \ll \{n_1, n_2\}$. The forward pass can be denoted as

$$h = W_0 x + \Delta W x = W_0 x + BA x. \tag{8}$$

## 3.2 LORA DROPOUT

The conventional dropout method only conducts dropout on the hidden representations, which is equivalent to masking random columns for the delta weight matrix $\Delta W$. Such dropping strategy fails to generate sufficiently diverse sparsity patterns. However, it is also not realistic to generate a sparsity mask that covers every single element in $\Delta W$, since current LLMs usually consist a huge number of parameters, and generating such sparsity masks will leads to great GPU memory overhead. Therefore, we propose a dropout strategy in the reparameterized space of LoRA to improve the diversity of the generated sparsity pattern without significantly increasing GPU memory overhead. Specifically, for a LoRA module described in Eq.8, we randomly drop rows and columns from both tunable low-rank parameter matrices:

$$\hat{A} = A \cdot \text{diag}(m_A), m_A \sim \text{Bern}(1-p); \ \hat{B} = \left(B^\top \cdot \text{diag}(m_B)\right)^\top, m_B \sim \text{Bern}(1-p), \tag{9}$$

where $m_A \in \mathbb{R}^{n_2}$ and $m_B \in \mathbb{R}^{n_1}$ are mask vectors drawn from the Bernoulli distribution, and $p$ denotes the probability that the parameters get dropped. Note that we conduct dropout on the input/output dimension of both matrices as applying dropout on the rank dimension would decrease the rank of LoRA, significantly impacting its expressive power. Additionally, performing dropout on the rank dimension will not increase the sparsity of the product of LoRA matrices, while theoretical evidence in Section 2 highlights the significance of sparsity in our framework. With the dropout, the forward pass would be $\hat{h} = W_0 x + \hat{B}\hat{A} x$.

It should be noted that our dropout method is not only applicable to the original LoRA, but also equally suitable for LoRA-based variant methods, as long as they take the form of low-rank matrix decomposition. For example, AdaLoRA (Zhang et al., 2023) conducts the decomposition through a quasi-SVD method, $\Delta W = P \Lambda Q$, where $P \in \mathbb{R}^{n_1 \times r}$ and $Q \in \mathbb{R}^{r \times n_2}$ are left and right singular vectors, respectively, and $\Lambda \in \mathbb{R}^{r \times r}$ is a diagonal matrix containing singular values. It's easy to adapt our LoRA Dropout to AdaLoRA through:

$$\hat{P} = \left(P^\top \cdot \text{diag}(m_P)\right)^\top, m_P \sim \text{Bern}(1-p); \ \hat{Q} = Q \cdot \text{diag}(m_Q), m_Q \sim \text{Bern}(1-p). \tag{10}$$

Dropout is not applied on the $\Lambda$ matrix as it will also lead to rank shrinking and further influence the expressive power. Moreover, $\Lambda$ will be adjusted by the AdaLoRA algorithm by filtering out minor compositions in practice, hence we conduct no further dropout on it. After dropout, the delta weight matrix would be $\Delta \hat{W} = \hat{P} \Lambda \hat{Q}$. We provide a schematic diagram in Figure 2 illustrating the integration of the proposed LoRA Dropout framework with both LoRA and AdaLoRA methods.

**Training Objective** Let us denote $m$ as the concatenation of all dropout vectors from LoRA module of a fine-tuning model, $\Delta \theta(m)$ as the LoRA parameters after the dropout $m$, and $\theta^0$ as the

original pre-trained parameters. To obtain an effective model under various dropouts, we define the training objective as an average of multiple losses under $N$ different dropout instances on parameters,

$$\mathcal{L}(\boldsymbol{x}) = \frac{1}{N} \sum_{k=1}^{N} \ell\left(\boldsymbol{x}; \boldsymbol{\theta}^0 + \Delta\boldsymbol{\theta}(\boldsymbol{m}_k)\right), \boldsymbol{m}_k \sim \text{Bern}(1-p). \tag{11}$$

### 3.3 TEST-TIME ENSEMBLE STRATEGY

To further enhance the model's performance during inference time, inspired by the MC dropout mechanism (Gal & Ghahramani, 2016), we propose a test-time ensemble method. Unlike the conventional dropout that is deactivated when testing, our ensemble strategy aggregates the outputs of models under different dropouts during inference time to get the final output, which can be viewed as sampling and aggregating models from a parameter distribution with a Monte Carlo method. Specifically, let $\mathcal{M}\left(\boldsymbol{\theta}^0 + \Delta\boldsymbol{\theta}(\boldsymbol{m}_k)\right)$ denote the model with LoRA parameter under dropout $\boldsymbol{m}_k$, then the output $\boldsymbol{o}$ of the ensemble model is

$$\boldsymbol{o}(\boldsymbol{x}) = \frac{1}{N} \sum_{k=1}^{N} \boldsymbol{o}_k(\boldsymbol{x}) = \frac{1}{N} \sum_{k=1}^{N} \mathcal{M}\left(\boldsymbol{x}; \boldsymbol{\theta}^0 + \Delta\boldsymbol{\theta}(\boldsymbol{m}_k)\right), \tag{12}$$

where $N$ is the number of dropout instances. Here we justify the effectiveness of this test-time ensemble strategy by providing a theoretical analysis of how it helps further tighten the error bound.

During the fine-tuning phase, we optimize Eq.(3) through accumulating gradient steps under different dropout instances. This fine-tuning procedure is essentially optimizing the generalization risks given the distribution $\mathcal{D}$ of model parameters $\boldsymbol{\theta}$, which is $\mathbb{E}_{\boldsymbol{\theta} \sim \mathcal{D}} \mathbb{E}_{(x,y)} \mathcal{L}_\lambda(\mathcal{M}(x; \boldsymbol{\theta}), y)$, where $\mathcal{M}(x; \boldsymbol{\theta})$ denotes the output of model $\mathcal{M}$ given the input $x$ parameterized by $\boldsymbol{\theta}$. During the inference phase, with the test-time ensemble strategy, we are actually aggregating model outputs across the distribution $\mathcal{D}$ of parameter $\boldsymbol{\theta}$ to conduct final predictions, namely the ensemble classifier, which has an error of $\mathbb{E}_{(x,y)} \mathcal{L}_\lambda(\mathbb{E}_{\boldsymbol{\theta} \sim \mathcal{D}} \mathcal{M}(x; \boldsymbol{\theta}), y)$. We present the following proposition that depicts a tighter generalization error bound with the ensemble classifier:

**Proposition 3.1** (Error Bound of Test-time Ensemble). *If the loss function $\mathcal{L}_\lambda$ (e.g. cross-entropy) is convex w.r.t. the final output $\boldsymbol{o} = \mathcal{M}(x; \boldsymbol{\theta})$ of model $\mathcal{M}$, then we have:*

$$\mathbb{E}_{(x,y)} \mathcal{L}_\lambda(\mathbb{E}_{\boldsymbol{\theta} \sim \mathcal{D}} \mathcal{M}(x; \boldsymbol{\theta}), y) \leq \mathbb{E}_{\boldsymbol{\theta} \sim \mathcal{D}} \mathbb{E}_{(x,y)} \mathcal{L}_\lambda(\mathcal{M}(x; \boldsymbol{\theta}), y). \tag{13}$$

*Proof Sketch.* Taking expectation on parameter $\boldsymbol{\theta}$ is equivalent to taking expectation on the final output $\boldsymbol{o}$ from a certain distribution. Then simply apply Jensen inequality under the convex condition of $\boldsymbol{o}$ and the inequality holds. See Appendix A.2 for detailed proofs. $\square$

Moreover, the convexity holds for most cases in LLM training or fine-tuning scenarios, as we often take cross-entropy as the loss function and the softmax as the final output layer, and those functions are convex in the entire space $\mathbb{R}^d$. Hence, the inequality says that the generalization error of the ensemble classifier (i.e., LHS of (13)) is no greater than the training generalization error (i.e., RHS of (13)) for most LLM tuning scenarios, implying that the ensemble classifier with LoRA Dropout can further compress the error bound given by the LoRA Dropout fine-tuning and demonstrate better test-time generalizability.

To summarize, Theorem 2.4 and 3.1 together depict the full theoretical sketch of our practical framework. The fine-tuning phase applies LoRA Dropout to control the generalization error by balancing the trade-off between the empirical risk minimization and the complexity of adaptation, and the inference phase applies multiple dropout instances to accomplish an ensemble classifier with a tighter error bound and further enhances the test-time generalizability.

## 4 EXPERIMENTS

In this section, we conduct a series of experiments to validate the effectiveness of our proposed LoRA Dropout framework. We incorporate LoRA Dropout into LoRA-series works, LoRA and AdaLoRA, and compare them with original models (w/ and w/o conventional dropout strategy) and other baselines on various NLP tasks.

Table 1: Results with DeBERTaV3-base on GLUE. The results in **bold** indicate models with LoRA Dropout outperform their corresponding base models and other PEFT baselines. Results are averaged over 5 runs using different random seeds, and † indicates the p-value of the t-test is below 0.01.

| Method | | MNLI M-Acc | SST-2 Acc | CoLA Mcc | QQP Acc | QNLI Acc | RTE Acc | MRPC Acc | STS-B Corr | All Avg. |
|---|---|---|---|---|---|---|---|---|---|---|
| Full Fine-Tuning | | 89.90 | 95.63 | 69.19 | 92.40 | 94.03 | 83.75 | 89.46 | 91.60 | 88.25 |
| BitFit | | 89.37 | 94.84 | 66.96 | 88.41 | 92.24 | 78.70 | 87.75 | 91.35 | 86.20 |
| H-Adapter | | 90.13 | 95.53 | 66.64 | 91.91 | 94.11 | 84.48 | 89.95 | 91.48 | 88.28 |
| P-Adapter | | 90.33 | 95.61 | 68.77 | 92.04 | 94.29 | 85.20 | 89.46 | 91.54 | 88.41 |
| LoRA | original | 90.65 | 94.95 | 69.82 | 91.99 | 93.87 | 85.20 | 89.95 | 91.60 | 88.50 |
| | w/ dropout | 90.07 | 94.26 | 70.87 | 91.66 | 94.44 | 86.64 | 90.20 | 91.60 | 88.72 |
| | w/ LoRA Dropout | **90.85**† | **95.87**† | **71.32**† | **92.22**† | **94.56** | **88.09**† | **91.42**† | **92.00**† | **89.54**† |
| AdaLoRA | AdaLoRA | 90.76 | 96.10 | 71.45 | 92.23 | 94.55 | 88.09 | 90.69 | 91.84 | 89.46 |
| | w/ dropout | 90.49 | 95.99 | 71.20 | 91.45 | 94.56 | 87.36 | 90.20 | 91.75 | 89.13 |
| | w/ LoRA Dropout | 90.75 | **96.22** | **72.04**† | 92.04 | 94.47 | **88.81**† | **91.18**† | **92.07**† | **89.70**† |

**Baselines** We compared the our method with following state-of-the-art PEFT methods.

**(1) BitFit** (Zaken et al., 2022). Only the bias vectors from the model parameters get fine-tuned. **(2) H-Adapter** (Houlsby et al., 2019). The adapters are inserted between the MLP and the self-attention modules. **(3) P-Adapter** (Pfeiffer et al., 2020). Adapter layers are applied only after the MLP or the LayerNorm layer. **(4) LoRA** (Zhang et al., 2023). LoRA decomposes the learnable delta parameter matrix into two low-rank matrices to improve parameter efficiency. **(5) AdaLoRA** (Zhang et al., 2023). AdaLoRA introduces an adaptive parameter budget by gradually pruning the rank of LoRA based on sensitivity-based importance scores.

When comparing with baseline models, we keep tunable parameter budgets for all methods aligned. We set the hyperparametersto be the same as our base models, i.e. LoRA and AdaLoRA, following (Zhang et al., 2023), and only tune the hyperparameters that are exclusive to our model. More detailed experiment settings can be viewed in Appendix C and D.

### 4.1 NATURAL LANGUAGE UNDERSTANDING

**Settings** Following previous work (Zhang et al., 2023), we use the General Language Understanding Evaluation (GLUE) benchmark (Wang et al., 2018) for evaluation. Our experiments contain eight different tasks from the GLUE benchmark. All models are fine-tuned on the DeBERTaV3-base (He et al., 2021) pre-trained model.

**Results** The results of different models on the GLUE benchmark are shown in Table 1. From the results, we could find that LoRA-series models with our LoRA Dropout framework consistently outperform other baselines, and achieve the best performance on the overall result among eight NLU tasks. Moreover, when compared with the original LoRA and AdaLoRA models, models with our dropout method always achieve superior performance, indicating that the proposed LoRA Dropout framework could help LoRA-based models improve generalization ability on downstream tasks.

We could also find that LoRA models with our method achieve better results than the conventional dropout, which indicates that our dropout strategy could generate more diverse sparsity patterns and lead to better overfitting reduction. It is worth noting that LoRA model with conventional dropout sometimes performs worse that the original model. A possible explanation is that the rigid sparsity pattern (i.e., dropout columns only) constrains the learning dynamics of parameters and prevents the model from adequately exploring diverse feature combinations, leading to suboptimal fitting.

We also provide the loss curves of LoRA and AdaLoRA with LoRA dropout during fine-tuning on the RTE dataset in Figure 1. We could find that the gaps between curves on train and test set get significantly narrowed compared with the model without LoRA Dropout. That demonstrates our method's ability to reduce overfitting during fine-tuning. Moreover, we show that our LoRA Dropout method could help improve model calibration. The experiments and analyses can be found in Appendix E.

Table 2: Results with DeBERTaV3-base on SQuAD v1.1 and SQuADv2.0. We report EM and F1 for each model. The results in **bold** indicate models with LoRA Dropout outperform their corresponding base models and other PEFT baselines. Results are averaged over 5 runs using different random seeds, and † indicates the p-value of the t-test is below 0.01.

| | | SQUAD v1.1 | | | | | | SQUAD v2.0 | | | | |
|---|---|---|---|---|---|---|---|---|---|---|---|---|
| **#Param ratio** | | 0.16% | | 0.32% | | 0.65% | | 0.16% | | 0.32% | | 0.65% |
| **Metric** | | EM | F1 | EM | F1 | EM | F1 | EM | F1 | EM | F1 | EM | F1 |
| | H-Adapter | 85.3 | 92.1 | 86.1 | 92.7 | 86.7 | 92.9 | 84.3 | 87.3 | 84.9 | 87.9 | 85.4 | 88.3 |
| | P-Adapter | 85.9 | 92.5 | 86.2 | 92.8 | 86.6 | 93.0 | 84.5 | 87.6 | 84.9 | 87.8 | 84.5 | 87.5 |
| LoRA | original | 86.6 | 92.9 | 86.7 | 93.1 | 86.7 | 93.1 | 83.6 | 86.7 | 84.5 | 87.4 | 85.0 | 88.0 |
| | w/ dropout | 87.7 | 93.5 | 87.7 | 93.7 | 87.4 | 93.1 | 84.0 | 87.1 | 84.7 | 87.5 | 85.3 | 88.1 |
| | w/ LoRA Dropout | **88.2**† | **93.8**† | **88.7**† | **94.1**† | **88.7**† | **94.2**† | **85.4**† | **88.4**† | **86.0**† | **88.8**† | **86.1**† | **88.9**† |
| AdaLoRA | original | 87.5 | 93.6 | 87.5 | 93.7 | 87.6 | 93.7 | 85.7 | 88.8 | 85.5 | 88.6 | 86.0 | 88.9 |
| | w/ dropout | 88.2 | 94.1 | 88.1 | 94.0 | 88.2 | 94.1 | 85.8 | 88.7 | 85.7 | 88.7 | 85.8 | 88.8 |
| | w/ LoRA Dropout | 88.1 | 93.9 | **88.5**† | **94.2**† | **88.7**† | **94.3**† | 85.8 | 88.6 | **85.9**† | **88.9**† | **86.3**† | **89.1**† |

Table 3: Results of instruction tuning on LLaMA2-7B. We report Accuracy(%) for MMLU and average GPT-4-turbo score for Vicuna-Eval. The Best results are in **bold**.

| Method | MMLU (5-shot) | | | | | MMLU (0-shot) | | | | | Vicuna-Eval |
|---|---|---|---|---|---|---|---|---|---|---|---|
| | STEM | Social | Hum. | Other. | Avg. | STEM | Social | Hum. | Other. | Avg. | Score |
| LLaMA2-7B | 36.80 | **51.42** | 42.76 | 52.10 | 45.49 | 33.31 | 46.78 | 38.76 | 45.04 | 40.79 | 2.66 |
| LoRA$_{r=16}$ | **37.53** | 50.93 | 42.33 | 52.16 | 45.68 | **34.40** | 45.15 | 38.19 | 45.60 | 40.61 | 5.29 |
| LoRA Dropout | 36.50 | 51.13 | **43.56** | **52.88** | **45.86** | 34.13 | **48.74** | **40.47** | **47.43** | **42.53** | **6.03** |

## 4.2 QUESTION ANSWERING

**Settings** We conduct the question answering task on two SQuAD (Stanford Question Answering Dataset) benchmarks, SQuAD v1.1 (Rajpurkar et al., 2016) and SQuAD v2.0 (Rajpurkar et al., 2018), with DeBERTaV3-base as the base pre-trained model. We report the Exact Match accuracy and F1 score for each method.

**Results** The results of different models on the SQuAD benchmarks are shown in Table 2. The results further validate the conclusions that we obtained from Table 1. The LoRA Dropout method helps the base model (i.e., LoRA and AdaLoRA) to achieve better performances on both SQuAD benchmarks. Moreover, by varying the budget of trainable parameters (i.e., the hidden dimension of adapters and the rank of LoRA module), we could find that our method has consistently superior performance under various parameter budgets, revealing its effectiveness.

## 4.3 INSTRUCTION TUNING

**Settings** We evaluate the models' natural language generation ability by conducting instruction tuning. Specifically, we choose LLaMA2-7B (Touvron et al., 2023) as the pre-trained base model and fine-tune on the Alpaca-clean dataset[1] (Taori et al., 2023) with original LoRA method and LoRA Dropout. We employ the MMLU benchmark (Hendrycks et al., 2021) and the Vicuna-Eval (Chiang et al., 2023) to evaluate each model. MMLU requires the models to answer multiple-choice tasks from different domains, and Vicuna-Eval prompts the model to respond to 80 predefined questions and utilizes the GPT-4 (Achiam et al., 2023) model to assess the answer qualities.

**Results** We report the results of the MMLU benchmark and Vicuna-Eval in Table 3, and provide answers to several different Vicuna-Eval questions generated by different models in Appendix H. All results show that our LoRA Dropout model achieves better performances on both benchmarks than the original LLaMA model and LoRA-finetuned model. With our dropout framework, the generalization ability of the fine-tuned model gets improved, leading to a better ability to apply the knowledge from the fine-tuning dataset to natural language response tasks.

## 4.4 ABLATION STUDIES AND SENSITIVITY ANALYSIS

---

[1]https://huggingface.co/datasets/yahma/alpaca-cleaned

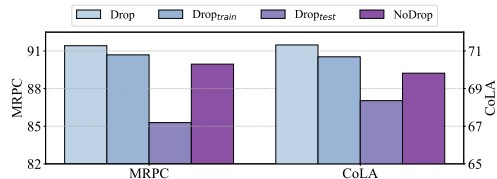

**Effect of Train/Test Dropout** We conduct experiments on the effects of dropout during training and testing on the MRPC and CoLA dataset, and the results are shown in Figure 3. We compared our method (LoRA with our LoRA Dropout framework, denoted as *Drop*) with the following variants: $Drop_{train}$ denotes training with dropout and testing without dropout ensemble. $Drop_{test}$ denotes training without dropout

Figure 3: Ablation studies on the dropout strategy.

and testing with dropout ensemble. And *NoDrop* denotes model without LoRA Dropout. We could find the importance of conducting dropout during fine-tuning from the bad performance of $Drop_{test}$. It performs worse than the vanilla *NoDrop* model since testing dropout may break some hidden semantic structure of parameters learned from training. We can also verify the effectiveness of test-time ensemble strategy from the decrease between *Drop* and $Drop_{train}$, which aligns with our theoretical derivation that the ensemble strategy would further compress the error bound.

**Effect of Dropout Rate** We conduct experiments on the effects of dropout rate $p$ on the MRPC dataset, and show the results in Figure 4. As the dropout rate increases, the performance first improves and then drops. This aligns with our theoretical derivation in Section 2.2 that a proper dropout rate would help balance the empirical risk minimization and complexity of the adaptation

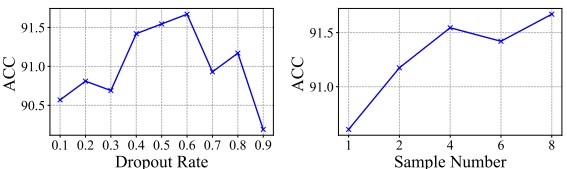

Figure 4: Results of the sensitivity analyses.

function. A small dropout rate might fail to introduce sufficient sparsity and lead to overfitting, while an excessively large dropout rate would result in too few trainable parameters, making the adapter lose its expressive power.

**Effect of Number of Sampled Dropout Instances** We conduct experiments on the effects of dropout instance number $N$ on the MRPC dataset, and the results are shown in Figure 4. From the results, we can find the model performance improves as the sample number increases. This is reasonable since with a larger sample number, more dropout instances can be introduced during training and more models get aggregated during the test-time ensemble, leading to more accurate estimations of the outputs over the parameter distribution. However, a larger sample number will also lead to higher training and inference costs, thus picking an appropriate $N$ is necessary for a better balance between accuracy and computational cost.

**Discussion - Varying LoRA Rank v.s. LoRA Dropout** In this subsection, we attempt to discuss whether simply shrinking LoRA rank can also reduce the risk of overfitting and achieve a better performance than LoRA Dropout. Here we provide the performances of LoRA models with various ranks (i.e., $r = \{2, 4, 8, 16\}$) and compare them with LoRA Dropout with a fixed rank (i.e., $r = 8$). The results are illustrated in Figure 5. Results show that the performance of LoRA gets better when shrinking the rank to 4, but dramatically decreases when shrunk to 2. This demonstrates that an excessively small

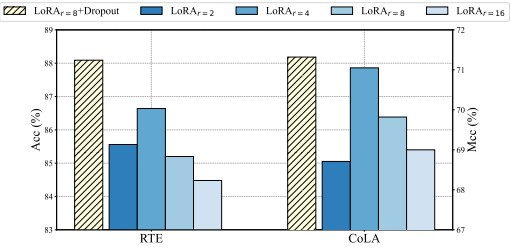

Figure 5: Comparisons between LoRA Dropout and LoRA with different ranks.

rank would limit the expressive power of LoRA and lead to huge performance decay. However, LoRA$_{r=4}$ still cannot outperform LoRA$_{r=8}$+Dropout with a dropout rate of 0.5. Though the activated parameters are exactly the same under both scenarios, LoRA Dropout enjoys better expressiveness (i.e., rank) without a higher overfitting risk, which can potentially learn better patterns than simply halving the rank.

Moreover, further enlarging the rank (i.e., $r = 16$) will deteriorate the LoRA performance due to the increasing risk of overfitting, as discussed in the Introduction. Therefore, we summarize that the effectiveness of LoRA Dropout comes from the capability of reducing overfitting while maintaining a stronger expressiveness.

## 5 RELATED WORKS

**Parameter-Efficient Fine-Tuning (PEFT)**   PEFT focuses on efficiently adapting pre-trained language models to downstream tasks by fine-tuning a few additional parameters or a subset of pre-trained parameters. Current mainstream PEFT approach can be roughly divided into three categories (Lialin et al., 2023; Xu et al., 2023). *Additive Fine-tuning* methods (Houlsby et al., 2019; Pfeiffer et al., 2020; He et al., 2022; Lester et al., 2021; Li & Liang, 2021) focus on adding extra tunable parameters by introducing additional layers or learnable prompts. *Partial Fine-tuning* methods (Zaken et al., 2022; Xu et al., 2021; Fu et al., 2023) select a subset of pre-trained parameters for fine-tuning. *Reparameterization Fine-tuning* methods (Hu et al., 2021; Zhang et al., 2023; Edalati et al., 2022) adopt low-rank representations to minimize the number of trainable parameters. In this paper, we focus on the most effective and widely adopted method, LoRA (Hu et al., 2021) and its variants, which decompose the learnable delta weight into the product of two low-rank matrices. The rank of the decomposition is essential for LoRA. A small rank may lead to insufficient expressive power, while a large rank could result in overfitting. One of LoRA's variants, AdaLoRA (Zhang et al., 2023) proposes to decompose the delta weight through a quasi-SVD method, and select parameters through importance scoring. Nevertheless, this selection method also relies on gradients on the training set, leading to an additional risk of overfitting. In this work, we propose a theoretically grounded dropout framework for LoRA-series methods, filling the gap that the LoRA-based PEFT methods lack theoretical guidances and practical mechanisms to control overfitting.

**Dropout Regularization**   The dropout mechanism (Hinton et al., 2012) is a well-known and widely-adopted technique in deep neural networks to prevent overfitting. In standard dropout, each neuron in the network is omitted from the network with a certain possibility during training. Subsequently, various dropout techniques for specific model structures are introduced, like Spatial dropout (Tompson et al., 2015) for convolutional layers and Recurrent dropout (Semeniuta et al., 2016) for recurrent neural networks. Meanwhile, works have been done to explore the theoretical factors behind dropout's ability to suppress overfitting. Some works believe that the the model learns a geometric mean over the ensemble of possible sub-networks through dropout (Warde-Farley et al., 2013; Baldi & Sadowski, 2013), and some works view dropout from a Bayesian perspective and argue that model with dropout can be interpreted as a Bayesian model approximating a posterior over parameters (Gal & Ghahramani, 2016). Recently there has been work trying to combine dropout with LLM (Wang et al., 2024). But this work mainly focuses on conduct dropout on the transformer structure instead of LoRA, and also lacks theoretical analysis. To the best of our knowledge, currently there's little theoretical work on applying dropout on LoRA-based PEFT models, where fine-tuning happens on the delta weight matrices with low-rank decompositions.

## 6 CONCLUSIONS AND LIMITATIONS

To control the overfitting risk when fine-tuning on downstream tasks, in this paper, we propose a theoretically grounded LoRA Dropout framework designed for LoRA-based PEFT methods. Theoretical analyses from the perspective of sparse show that sparsity introduced by LoRA Dropout helps tighten the between empirical and generalization risks and thereby control overfitting. A test-ensemble strategy is proposed based on LoRA Dropout and theoretically shown to further compress the error bound. We conduct experiments on various tasks and PLMs, and the results demonstrate the effectiveness of our method on improving model'performances.

Despite the promising results, we still want to point out the limitations. Though LoRA Dropout introduces no additional tunable parameters compared to LoRA, sampling multiple dropout instances during training and testing does introduce considerable time overhead. For future work, we aim to design a parallel computing framework for LoRA Dropout, expecting to improve in both performance and efficiency.

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

# A    PROOFS OF THEORETICAL RESULTS

## A.1    PROOF OF PROPOSITION 2.2

We first prove a Lemma that describes the pointwise hypothesis stability of an optimization problem with $\ell_2$-regularizer, which provides an upper bound related to a constant describing the specific shape around the local optima. The symbols follow those in the main text. We first denote $\mathcal{L}(\boldsymbol{\theta}) = \frac{1}{n}\sum_i \mathcal{L}(x_i; \boldsymbol{\theta})$.

**Lemma A.1.** *Consider the learning algorithm $\mathcal{M}$ optimizing the following loss function:*

$$\min_{\theta} \mathcal{L}_\lambda(\boldsymbol{\theta}) := \min_{\boldsymbol{\theta}} \mathcal{L}(\boldsymbol{\theta}) + \lambda ||\boldsymbol{\theta} - \boldsymbol{\theta}^0||_2^2.$$

*If the loss function $\mathcal{L}$ is $\eta$-Lipschitz, $\boldsymbol{\theta}_{\mathcal{L}_\lambda}(\mathbf{S})$ is close to $\boldsymbol{\theta}_{\mathcal{L}_\lambda}(\mathbf{S}^i)$ whose gap is bounded by a small constant $\epsilon \to 0$, i.e., $||\boldsymbol{\theta}_{\mathcal{L}_\lambda}(\mathbf{S}) - \boldsymbol{\theta}_{\mathcal{L}_\lambda}(\mathbf{S}^i)|| \leq \epsilon \to 0$, and the regularization coefficient $\lambda \geq \max\{0, \frac{\eta}{n\epsilon} - \frac{1}{2}\Lambda_{\min}\}$, where $\Lambda_{\min}$ is the minimum eigenvalue of the Hessian $\nabla^2\mathcal{L}(\boldsymbol{\theta}_{\mathcal{L}_\lambda}(\mathbf{S}))$ at $\boldsymbol{\theta}_{\mathcal{L}_\lambda}(\mathbf{S})$, unitary diagonalized as $U\operatorname{diag}(\Lambda)U^{-1}, \Lambda = \{\Lambda_1, \cdots, \Lambda_d\}$ and $\Lambda_{\min} = \min\{\Lambda_1, \cdots, \Lambda_d\}$, Then $\mathcal{M}$ has pointwise hypothesis stability $\beta = \frac{2\eta^2}{(\Lambda_{\min}+2\lambda)n}$, which is:*

$$\mathbb{E}_{\mathbf{S},i\sim\mathrm{U}(n)} \left| \mathcal{L}_\lambda\left(x_i; \boldsymbol{\theta}_{\mathcal{L}_\lambda}(\mathbf{S}^i)\right) - \mathcal{L}_\lambda\left(x_i; \boldsymbol{\theta}_{\mathcal{L}_\lambda}(\mathbf{S})\right) \right| \leq \frac{2\eta^2}{(\Lambda_{\min} + 2\lambda)\, n}.$$

*Proof.* For simplicity, we denote $\boldsymbol{\theta}_{\mathcal{L}_\lambda}(\mathbf{S})$ as $\hat{\boldsymbol{\theta}}$, and $\Delta\hat{\boldsymbol{\theta}} := \hat{\boldsymbol{\theta}} - \boldsymbol{\theta}$. Consider the second-order Taylor expansion of $\mathcal{L}_\lambda$ at local optima $\hat{\boldsymbol{\theta}}$, we have $\nabla_{\mathcal{L}_\lambda}(\hat{\boldsymbol{\theta}}) = 0$. For $\forall v$ close to $\hat{\boldsymbol{\theta}}$, we have:

$$\mathcal{L}_\lambda(v) = \mathcal{L}_\lambda(\hat{\boldsymbol{\theta}}) + (v - \hat{\boldsymbol{\theta}})^\top \nabla_{\mathcal{L}_\lambda}(\hat{\boldsymbol{\theta}}) + \frac{1}{2}(v - \hat{\boldsymbol{\theta}})^\top \nabla^2\mathcal{L}_\lambda(\hat{\boldsymbol{\theta}})(v - \hat{\boldsymbol{\theta}})$$

$$= \mathcal{L}_\lambda(\hat{\boldsymbol{\theta}}) + \frac{1}{2}(v - \hat{\boldsymbol{\theta}})^\top \nabla^2\mathcal{L}_\lambda(\hat{\boldsymbol{\theta}})(v - \hat{\boldsymbol{\theta}})$$

Then, we have:

$$\mathcal{L}_\lambda(v) - \mathcal{L}_\lambda(\hat{\boldsymbol{\theta}})$$

$$= \frac{1}{2}(v - \hat{\boldsymbol{\theta}})^\top \nabla^2\mathcal{L}_\lambda(\hat{\boldsymbol{\theta}})(v - \hat{\boldsymbol{\theta}})$$

$$= \frac{1}{2}(v - \hat{\boldsymbol{\theta}})^\top \nabla_{\hat{\boldsymbol{\theta}}}^2(\mathcal{L}(\hat{\boldsymbol{\theta}}) + \lambda ||\hat{\boldsymbol{\theta}} - \boldsymbol{\theta}^0||_2^2)(v - \hat{\boldsymbol{\theta}})$$

$$= \frac{1}{2}(v - \hat{\boldsymbol{\theta}})^\top (\nabla^2\mathcal{L}(\hat{\boldsymbol{\theta}}) + 2\lambda I)(v - \hat{\boldsymbol{\theta}})$$

$$= \frac{1}{2}(v - \hat{\boldsymbol{\theta}})^\top (U\operatorname{diag}(\Lambda)U^{-1} + 2\lambda I)(v - \hat{\boldsymbol{\theta}})$$

$$= \frac{1}{2}(v - \hat{\boldsymbol{\theta}})^\top (U(\operatorname{diag}(\Lambda) + 2\lambda I)U^{-1})(v - \hat{\boldsymbol{\theta}})$$

$$= \frac{1}{2}(v - \hat{\boldsymbol{\theta}})^\top (U\operatorname{diag}(\sqrt{\Lambda_1 + 2\lambda}, \cdots, \sqrt{\Lambda_d + 2\lambda})U^{-1}U\operatorname{diag}(\sqrt{\Lambda_1 + 2\lambda}, \cdots, \sqrt{\Lambda_d + 2\lambda})U^{-1})(v - \hat{\boldsymbol{\theta}})$$

$$= \frac{1}{2}||(U\operatorname{diag}(\sqrt{\Lambda_1 + 2\lambda}, \cdots, \sqrt{\Lambda_d + 2\lambda}U^{-1})(v - \hat{\boldsymbol{\theta}})||_2^2$$

$$\geq \frac{1}{2}(\Lambda_{\min} + 2\lambda)||v - \hat{\boldsymbol{\theta}}||_2^2.$$

(A.1)

This inequality holds for the orthogonality of $U$ that it does not change the magnitude of vector $v - \hat{\boldsymbol{\theta}}$, and the magnitude is at least scaled with $\Lambda_{\min} + 2\lambda$. Then, by the definition of $\mathcal{L}_\lambda(\theta)$, for $\forall u, v$ close to $\hat{\boldsymbol{\theta}}$, we have:

$$\mathcal{L}_\lambda(u) - \mathcal{L}_\lambda(v)$$

$$= \left( \frac{1}{n} \sum_k \mathcal{L}(x_k; u) + \lambda ||u - \boldsymbol{\theta}^0||_2^2 \right) - \left( \frac{1}{n} \sum_k \mathcal{L}(x_k; v) + \lambda ||v - \boldsymbol{\theta}^0||_2^2 \right)$$

$$= \left( \frac{1}{n} \sum_{k \neq i} \mathcal{L}(x_k; u) + \lambda ||u - \boldsymbol{\theta}^0||_2^2 \right) - \left( \frac{1}{n} \sum_{k \neq i} \mathcal{L}(x_k; v) + \lambda ||v - \boldsymbol{\theta}^0||_2^2 \right) + \frac{\mathcal{L}(x_i; u) - \mathcal{L}(x_i; v)}{n}$$

$$= (1 - \frac{1}{n}) \left( \frac{1}{n-1} \sum_{k \neq i} \mathcal{L}(x_k; u) + \lambda ||u - \boldsymbol{\theta}^0||_2^2 \right) - (1 - \frac{1}{n}) \left( \frac{1}{n-1} \sum_{k \neq i} \mathcal{L}(x_k; v) + \lambda ||v - \boldsymbol{\theta}^0||_2^2 \right) +$$

$$\frac{\lambda \left( ||u - \boldsymbol{\theta}^0||_2^2 - ||v - \boldsymbol{\theta}^0||_2^2 \right)}{n} + \frac{\mathcal{L}(x_i; u) - \mathcal{L}(x_i; v)}{n}$$

$$= (1 - \frac{1}{n}) \underbrace{\left[ \left( \frac{1}{n-1} \sum_{k \neq i} \mathcal{L}(x_k; u) + \lambda ||u - \boldsymbol{\theta}^0||_2^2 \right) - \left( \frac{1}{n-1} \sum_{k \neq i} \mathcal{L}(x_k; v) + \lambda ||v - \boldsymbol{\theta}^0||_2^2 \right) \right]}_{(*)} +$$

$$\frac{\mathcal{L}_\lambda(x_i; u) - \mathcal{L}_\lambda(x_i; v)}{n}. \tag{A.2}$$

Taking $u = \boldsymbol{\theta}_{\mathcal{L}_\lambda}(\mathbf{S}^i)$ and $v = \boldsymbol{\theta}_{\mathcal{L}_\lambda}(\mathbf{S})$. As $u$ minimizes the empirical loss of removing $x_i$ out, hence the $(*)$ item in Eq.(A.2) is smaller than 0. Then, we have:

$$\mathcal{L}_\lambda(\boldsymbol{\theta}_{\mathcal{L}_\lambda}(\mathbf{S}^i)) - \mathcal{L}_\lambda(\boldsymbol{\theta}_{\mathcal{L}_\lambda}(\mathbf{S})) \leq \frac{\mathcal{L}_\lambda(x_i; \boldsymbol{\theta}_{\mathcal{L}_\lambda}(\mathbf{S}^i)) - \mathcal{L}_\lambda(x_i; \boldsymbol{\theta}_{\mathcal{L}_\lambda}(\mathbf{S}^i))}{n}$$

Considering inequality of (A.1), we have:

$$\frac{1}{2}(\Lambda_{\min} + 2\lambda)||\boldsymbol{\theta}_{\mathcal{L}_\lambda}(\mathbf{S}^i) - \boldsymbol{\theta}_{\mathcal{L}_\lambda}(\mathbf{S})||_2^2 \leq \frac{\mathcal{L}_\lambda(x_i; \boldsymbol{\theta}_{\mathcal{L}_\lambda}(\mathbf{S}^i)) - \mathcal{L}_\lambda(x_i; \boldsymbol{\theta}_{\mathcal{L}_\lambda}(\mathbf{S}))}{n} \tag{A.3}$$

As the loss function $\mathcal{L}_\lambda$ is $\eta$-Lipschitz, thus we have:

$$|\mathcal{L}_\lambda(x_i; \boldsymbol{\theta}_{\mathcal{L}_\lambda}(\mathbf{S}^i)) - \mathcal{L}_\lambda(x_i; \boldsymbol{\theta}_{\mathcal{L}_\lambda}(\mathbf{S}))| \leq \eta ||\boldsymbol{\theta}_{\mathcal{L}_\lambda}(\mathbf{S}^i) - \boldsymbol{\theta}_{\mathcal{L}_\lambda}(\mathbf{S})||. \tag{A.4}$$

Taking (A.4) into (A.3), we have:

$$\frac{1}{2}(\Lambda_{\min} + 2\lambda)||\boldsymbol{\theta}_{\mathcal{L}_\lambda}(\mathbf{S}^i) - \boldsymbol{\theta}_{\mathcal{L}_\lambda}(\mathbf{S})||_2^2 \leq \frac{\eta ||\boldsymbol{\theta}_{\mathcal{L}_\lambda}(\mathbf{S}^i) - \boldsymbol{\theta}_{\mathcal{L}_\lambda}(\mathbf{S})||}{n}$$

$$\Rightarrow ||\boldsymbol{\theta}_{\mathcal{L}_\lambda}(\mathbf{S}^i) - \boldsymbol{\theta}_{\mathcal{L}_\lambda}(\mathbf{S})|| \leq \frac{2\eta}{(\Lambda_{\min} + 2\lambda)n}. \tag{A.5}$$

Plugging (A.5) back to (A.4):

$$|\mathcal{L}_\lambda(x_i; \boldsymbol{\theta}_{\mathcal{L}_\lambda}(\mathbf{S}^i)) - \mathcal{L}_\lambda(x_i; \boldsymbol{\theta}_{\mathcal{L}_\lambda}(\mathbf{S}))| \leq \frac{2\eta^2}{(\Lambda_{\min} + 2\lambda)n}. \tag{A.6}$$

As this holds for any $i$ and $\mathbf{S}$, hence we have:

$$\mathbb{E}_{\mathbf{S}, i \sim \mathrm{U}(n)} \left| \mathcal{L}_\lambda \left( x_i; \boldsymbol{\theta}_{\mathcal{L}_\lambda}(\mathbf{S}^i) \right) - \mathcal{L}_\lambda \left( x_i; \boldsymbol{\theta}_{\mathcal{L}_\lambda}(\mathbf{S}) \right) \right| \leq \frac{2\eta^2}{(\Lambda_{\min} + 2\lambda) n}. \tag{A.7}$$

Taking the condition of $\lambda \geq \max\{0, \frac{\eta}{n\epsilon} - \frac{1}{2}\Lambda_{\min}\}$ into inequality (A.7), we can obtain that $\mathbb{E}_{\mathbf{S}, i \sim \mathrm{U}(n)} \left| \mathcal{L}_\lambda \left( x_i; \boldsymbol{\theta}_{\mathcal{L}_\lambda}(\mathbf{S}^i) \right) - \mathcal{L}_\lambda \left( x_i; \boldsymbol{\theta}_{\mathcal{L}_\lambda}(\mathbf{S}) \right) \right| \leq \frac{2\eta^2}{(\Lambda_{\min} + 2\lambda)n} \leq \epsilon\eta$. This denotes that (A.7) provides a tighter upper bound than that depicted by the $\eta$-Lipschitzness and $\epsilon$-closeness when the regularization strength is sufficiently large. In practice, we also apply the hard dropout mechanism to satisfy this condition. $\square$

Based on this Lemma, we aim to analyze our optimization objective of Eq.(3) and prove Proposition 2.2 as follows.

**Proposition A.2** (PHS Upper Bound of LoRA Fine-tuning with Dropout). *If the loss function $\mathcal{L}_\lambda$ of the algorithm $\mathcal{M}$ is $\eta$-Lipschitz, $\boldsymbol{\theta}_{\mathcal{L}_\lambda}(\mathbf{S})$ is close to $\boldsymbol{\theta}_{\mathcal{L}_\lambda}(\mathbf{S}^i)$ whose gap is bounded by a small constant $\epsilon \to 0$, i.e., $\|\boldsymbol{\theta}_{\mathcal{L}_\lambda}(\mathbf{S}) - \boldsymbol{\theta}_{\mathcal{L}_\lambda}(\mathbf{S}^i)\| \leq \epsilon \to 0$, and the regularization co-efficient $\lambda \geq \max\{0, \frac{\eta}{pn\epsilon} - \frac{1}{2p}\Lambda_{\min}\}$, where $\Lambda_{\min}$ is the minimum eigenvalue of the Hessian $\nabla^2\mathcal{L}(\boldsymbol{\theta}_{\mathcal{L}_\lambda}(\mathbf{S}))$ at $\boldsymbol{\theta}_{\mathcal{L}_\lambda}(\mathbf{S})$, unitary diagonalized as $U\operatorname{diag}(\Lambda)U^{-1}, \Lambda = \{\Lambda_1, \cdots, \Lambda_d\}$ and $\Lambda_{\min} = \min\{\Lambda_1, \cdots, \Lambda_d\}$, then the algorithm optimizing $\mathcal{L}_\lambda$ on $\mathbf{S}$ has an upper bound of pointwise hypothesis stability of:*

$$\mathbb{E}_{\mathbf{S}, i\sim \mathrm{U}(n)}\left|\mathcal{L}_\lambda\left(x_i; \boldsymbol{\theta}_{\mathcal{L}_\lambda}(\mathbf{S}^i)\right) - \mathcal{L}_\lambda\left(x_i; \boldsymbol{\theta}_{\mathcal{L}_\lambda}(\mathbf{S})\right)\right| \leq \frac{2\eta^2}{(\Lambda_{\min} + 2\lambda p)\,n}.$$

*Proof.* Consider loss function with sparsity regularization from Eq.(3), and we have:

$$\begin{aligned}
\mathcal{L}_\lambda(\boldsymbol{\theta}) &= \mathcal{L}(\boldsymbol{\theta}) + \lambda\mathbb{E}_{\boldsymbol{d}\sim\mathrm{Bern}(p)}\|\boldsymbol{d}\odot(\boldsymbol{\theta}-\boldsymbol{\theta}^0)\|_2^2 \\
&= \mathcal{L}(\boldsymbol{\theta}) + \lambda\mathbb{E}_{\boldsymbol{d}\sim\mathrm{Bern}(p)}\sum_i d_i^2(\theta_i - \theta_i^0)^2 \\
&= \mathcal{L}(\boldsymbol{\theta}) + \lambda\sum_i (\theta_i - \theta_i^0)^2\mathbb{E}_{d_i\sim\mathrm{Bern}(p)}d_i^2 \\
&= \mathcal{L}(\boldsymbol{\theta}) + \lambda\sum_i (\theta_i - \theta_i^0)^2 p \\
&= \mathcal{L}(\boldsymbol{\theta}) + \lambda p\|\boldsymbol{\theta} - \boldsymbol{\theta}^0\|_2^2.
\end{aligned}$$

Through taking the results above to Lemma A.1, we can substitute the regularization coefficient with $\lambda p$ and obtain the pointwise hypothesis stability of the algorithm with dropout, which is:

$$\mathbb{E}_{\mathbf{S}, i\sim \mathrm{U}(n)}\left|\mathcal{L}_\lambda\left(x_i; \boldsymbol{\theta}_{\mathcal{L}_\lambda}(\mathbf{S}^i)\right) - \mathcal{L}_\lambda\left(x_i; \boldsymbol{\theta}_{\mathcal{L}_\lambda}(\mathbf{S})\right)\right| \leq \frac{2\eta^2}{(\Lambda_{\min} + 2\lambda p)\,n}.$$

$\square$

## A.2  PROOF OF PROPOSITION 3.1

**Proposition A.3** (Error Bound of Test-time Ensemble). *If the loss function $\mathcal{L}_\lambda$ (e.g. cross-entropy) is convex w.r.t. the final output $\boldsymbol{o} = \mathcal{M}(x; \boldsymbol{\theta})$ of model $\mathcal{M}$, then we have:*

$$\mathbb{E}_{(x,y)}\mathcal{L}_\lambda(\mathbb{E}_{\boldsymbol{\theta}\sim\mathcal{D}}\mathcal{M}(x; \boldsymbol{\theta}), y) \leq \mathbb{E}_{\boldsymbol{\theta}\sim\mathcal{D}}\mathbb{E}_{(x,y)}\mathcal{L}_\lambda(\mathcal{M}(x; \boldsymbol{\theta}), y). \tag{A.8}$$

*Proof.* According to Jensen inequality, for any distribution $\boldsymbol{O}$ of outputs $\boldsymbol{o}$, under the convexity assumption of function $\mathcal{L}_\lambda$ we have:

$$\mathcal{L}_\lambda(\mathbb{E}_{\boldsymbol{o}\sim\boldsymbol{O}}(\boldsymbol{o}), y) \leq \mathbb{E}_{\boldsymbol{o}\sim\boldsymbol{O}}\mathcal{L}_\lambda(\boldsymbol{o}, y). \tag{A.9}$$

Let us denote the distribution of $\boldsymbol{o}$ under a certain input $x$ of all parameters $\theta$ from distribution $\mathcal{D}$ as $\mathcal{O}(x) := \mathcal{M}(x; \mathcal{D})$. We have:

$$\mathbb{E}_{(x,y)}\mathcal{L}_\lambda\left(\mathbb{E}_{\boldsymbol{\theta}\sim\mathcal{D}}\mathcal{M}(x; \boldsymbol{\theta}), y\right) = \mathbb{E}_{(x,y)}\mathcal{L}_\lambda\left(\mathbb{E}_{\boldsymbol{o}\sim\mathcal{O}(x)}(\boldsymbol{o}), y\right) \tag{A.10}$$

As inequality (A.9) holds for any distribution of $\boldsymbol{o}$, we have:

$$\begin{aligned}
\mathbb{E}_{(x,y)}\mathcal{L}_\lambda\left(\mathbb{E}_{\boldsymbol{o}\sim\mathcal{O}(x)}(\boldsymbol{o}), y\right) &\leq \mathbb{E}_{(x,y)}\mathbb{E}_{\boldsymbol{o}\sim\mathcal{O}(x)}\mathcal{L}_\lambda\left(\boldsymbol{o}, y\right) \\
&= \mathbb{E}_{(x,y)}\mathbb{E}_{\boldsymbol{\theta}\sim\mathcal{D}}\mathcal{L}_\lambda\left(\mathcal{M}(x; \boldsymbol{\theta}), y\right) \\
&= \mathbb{E}_{\boldsymbol{\theta}\sim\mathcal{D}}\mathbb{E}_{(x,y)}\mathcal{L}_\lambda\left(\mathcal{M}(x; \boldsymbol{\theta}), y\right).
\end{aligned} \tag{A.11}$$

Plugging (A.11) into (A.10) and the result is obtained. $\square$

## B  ALGORITHM

Here we provide an overall training and testing procedure for fine-tuning pre-trained models with the LoRA-based method and LoRA Dropout in Alg 1.

---

**Algorithm 1** The overall fine-tuning and testing procedure of a pre-trained model with LoRA Dropout.

1: **Input:** total epoch number $T$, batch size $B$, dropout rate $p$, dropout instance number $N$.
2: **Training Phase:**
3: **for** $epoch$ from 1 to $T$ **do**
4:     **for** each iteration **do**
5:         randomly draw $B$ samples from the training set;
6:         $\mathcal{L}_{tr} \leftarrow 0$;
7:         draw $\boldsymbol{m}_r \sim \text{Bern}(1-p)$, for $r$ in 1,...,$N$;
8:         **for** each sample $\boldsymbol{x}$ in batch **do**
9:             $\mathcal{L}_{tr} \leftarrow \mathcal{L}_{tr} + \mathcal{L}(\boldsymbol{x})$ by Eq.(11);
10:         **end for**
11:         update tunable parameters with $\nabla \mathcal{L}_{tr}$.
12:     **end for**
13: **end for**
14: **Test Phase:**
15: **for** each sample $\boldsymbol{x}$ in test set **do**
16:     draw $\boldsymbol{m}_r \sim \text{Bern}(1-p)$, for $r$ in 1,...,$N$;
17:     compute the ensemble output $\boldsymbol{o}(\boldsymbol{x})$ following Eq.(12).
18: **end for**

---

Table C.1: Summary of hyperparameter settings when fine-tuning on different tasks of the GLUE benchmark.

| Corpus | MNLI | RTE | QNLI | MRPC | QQP | SST-2 | CoLA | STS-B |
|---|---|---|---|---|---|---|---|---|
| learning rate | 5e-4 | 1.2e-3 | 5e-4 | 1e-3 | 5e-4 | 8e-4 | 5e-5 | 2.2e-3 |
| batch size | 32 | 32 | 32 | 32 | 32 | 32 | 32 | 32 |
| # epochs | 7 | 50 | 5 | 30 | 10 | 24 | 25 | 25 |
| dropout rate | 0.5 | 0.5 | 0.5 | 0.5 | 0.5 | 0.5 | 0.5 | 0.5 |
| sample number | 4 | 4 | 4 | 4 | 4 | 4 | 4 | 4 |

## C    EXPERIMENTAL DETAILS

### C.1    IMPLEMENTATION DETAILS OF NLU TASK

All of our experiments on NLU task are implemented based on PyTorch 1.9.1 with Python 3.7.16 on the HuggingFace transformers library (Wolf et al., 2019) 4.4.2. Fine-tuning is conducted on the pre-trained DeBERTaV3-base (He et al., 2021) model, and PEFT methods are applied on all the linear layers in every transformer block. We mainly follow the hyperparameter setting as (Zhang et al., 2023) and tune hyperparameters exclusive to our model. The hyperparameters used when fine-tuning on each BLUE task are shown in Table C.1. For the hardware environment, We perform our experiments on a single NVIDIA-A100-80GB GPU or distributedly on 2 NVIDIA-RTX3090-24GB GPUs. The approximate times for fine-tuning on each task in GLUE with 2 NVIDIA-RTX3090-24GB GPUs are shown in Table C.2.

### C.2    IMPLEMENTATION DETAILS OF QA TASK

All of our experiments on QA task are implemented based on PyTorch 1.9.1 with Python 3.7.16 on the HuggingFace transformers library (Wolf et al., 2019) 4.21.0. Fine-tuning is conducted on the pre-trained DeBERTaV3-base (He et al., 2021) model, and PEFT methods are applied on all the linear layers in every transformer block. We control the ratio of tunable parameters by adjusting the hyperparameters related to parameter budget, e.g. adapter dimension or LoRA rank. Specifically, the tunable parameter ratios of {0.16%,0.32%,0.65%} correspond to LoRA rank of {2,4,8} respectively. Other hyperparameters used when fine-tuning on SQuAD benchmark are shown in Table C.3. For the hardware environment, We perform our experiments on a single NVIDIA-A100-80GB GPU or distributedly on 2 NVIDIA-RTX3090-24GB GPUs. The time for fine-tuning on SQuAD v1.1 and

Table C.2: Approximate time to conduct experiments on different tasks of the GLUE benchmark with 2 NVIDIA-RTX3090-24GB GPUs

|  | MNLI | RTE | QNLI | MRPC | QQP | SST-2 | CoLA | STS-B |
|---|---|---|---|---|---|---|---|---|
| LoRA w/ dropout | 5 hrs | 15 mins | 1.5 hrs | 10 mins | 5 hrs | 2 hrs | 18 mins | 15 mins |
| LoRA w/ LoRA Dropout | 20 hrs | 40 mins | 5 hrs | 40 mins | 28 hrs | 8 hrs | 60 mins | 40 mins |
| AdaLoRA w/ dropout | 15 hrs | 25 mins | 4 hrs | 24 mins | 16 hrs | 7 hrs | 50 mins | 30 mins |
| AdaLoRA w/ LoRA Dropout | 28 hrs | 60 mins | 6 hrs | 60 mins | 30 hrs | 12 hrs | 100 mins | 75 mins |

SQuAD v2.0 on LoRA with dropout is 13 hours and 24 hours, respectively, and 15 hours and 28 hours for AdaLoRA with dropout.

Table C.3: Summary of hyperparameter settings when fine-tuning on the SQuAD benchmark.

| Corpus | SQuAD v1.1 | SQuAD v2.0 |
|---|---|---|
| learning rate | 1e-3 | 1e-3 |
| batch size | 16 | 16 |
| # epochs | 10 | 12 |
| dropout rate | 0.5 | 0.5 |
| sample number | 4 | 4 |

## C.3 IMPLEMENTATION DETAILS OF INSTRUCTION TUNING

When performing instruction tuning, we use PyTorch 2.1.2 with Python 3.10.13. We employ the PEFT library (Mangrulkar et al., 2022) and the LLaMA-Factory library (hiyouga, 2023) for implementing and evaluating our method. Fine-tuning is conducted on LLaMA2-7B (Touvron et al., 2023), and only the {q_proj,v_proj,k_proj,o_proj} linear modules in each transformer block get tuned. All hyperparameters used for fine-tuning LoRA and LoRA with dropout are shown in Table C.4. For the hardware environment, experiments are conducted distributedly on 2 NVIDIA-A100-80GB GPUs. It takes approximately 4 hours to fine-tune LoRA with dropout on the Alpaca-clean dataset.

# D DATASET DETAILS

## D.1 DETAILS OF GLUE BENCHMARK

We use the General Language Understanding Evaluation (GLUE) benchmark (Wang et al., 2018) for evaluation on NLU tasks. Following previous work (Zhang et al., 2023), eight datasets are picked for fine-tuning. Here we list detailed statistics of each dataset in Table D.1.

## D.2 DETAILS OF SQuAD BENCHMARK

The SQuAD (Stanford Question Answering Dataset) benchmark is a benchmark for question answering task collected from Wikipedia by crowd-workers. Specifically, the task is treated as a sequence labeling problem, where the probability of tokens from the start and end of the answer span are picked for prediction. SQuAD v1.1(Rajpurkar et al., 2016) is the first version of SQuAD, including over 100,000 question-answer pairs sourced from 536 articles. And SQuAD v2.0(Rajpurkar et al., 2018) adds 50,000 unanswerable questions written by humans based on SQuADv1.1. Therefore, SQuAD v2.0 further demands the model to be able to differentiate whether a question is unanswerable. Statistics of both SQuAD datasets are shown in Table D.2.

Table C.4: Summary of hyperparameter settings when fine-tuning LLaMA2-7B.

| Hyperparameter | lr | batch size | rank | lr-scheduler | warmup step | dropout rate | sample num |
|---|---|---|---|---|---|---|---|
| **LoRA Dropout** | 5e-5 | 128 | 16 | cosine | 500 | 0.5 | 4 |

Table D.1: Summary of dataset statistic of the GLUE benchmark.

| Corpus | Task | Task Category | #Train | #Dev | #Label | Metrics |
|---|---|---|---|---|---|---|
| **CoLA** | Acceptability | Single-Sentence Classification | 8.5k | 1k | 2 | Matthews Corr |
| **SST** | Sentiment | Single-Sentence Classification | 67k | 872 | 2 | Matched Accuracy |
| **MNLI** | NLI | Pairwise Text Classification | 393k | 20k | 3 | Accuracy |
| **RTE** | NLI | Pairwise Text Classification | 2.5k | 276 | 2 | Accuracy |
| **QQP** | Paraphrase | Pairwise Text Classification | 364k | 40k | 2 | Accuracy |
| **MRPC** | Paraphrase | Pairwise Text Classification | 3.7k | 408 | 2 | Accuracy |
| **QNLI** | QA/NLI | Pairwise Text Classification | 108k | 5.7k | 2 | Accuracy |
| **STS-B** | Similarity | Text Similarity | 7k | 1.5k | 1 | Pearson Corr |

### D.3   DETAILS OF ALPACA DATASET BENCHMARK

We fine-tune LLaMA2-7B on the Alpaca-clean dataset[2]. Alpaca-clean is the cleaned version of the original Alpaca dataset (Taori et al., 2023). It consists of 51K instructions and demonstrations and is suitable for instruction-tuning. The cleaned version fixed multiple issues in the original release, including hallucinations, merged instructions, empty outputs, empty code examples, and instructions to generate images.

## E   EXPERIMENTS ON CONFIDENCE CALIBRATION

**Settings**   As large-scale pre-trained models often exhibit overconfidence (Jiang et al., 2021; Xiao et al., 2022; He et al., 2023; Tian et al., 2023), we evaluate the confidence calibration (Guo et al., 2017) of each model, which serves as an effective analytical method for evaluating model reliability (Zhu et al., 2023). Specifically, we employ the Expected Calibration Error (ECE) for measuring the calibration performance, and assess the confidence calibration of different fine-tuned models on a few tasks from the GLUE benchmark based on the DeBERTaV3-base model.

**Results**   We report the ECE results of each model in Table E.1 when it reaches the best performance on the development set, and also provide the ECE curves of different models in Figure 1 when fine-tuned on the RTE task. From the results we could find that LoRA Dropout could consistently reduce the ECE compared with its base model, leading to better-calibrated models. One possible explanation is that LoRA Dropout can be viewed as a variant of the MC dropout from a Bayes perspective. By randomly dropping parameters during training, we are estimating the posterior weight distributions with a given downstream task, making the model a kind of Bayes neural network, which is known to achieve good calibration (Kristiadi et al., 2020).

## F   EXPERIMENTS ON VISUAL TASKS

**Settings**   Except for NLP tasks, we also conduct experiments on visual tasks. Here we use the VTAB-1K benchmark  (Zhai et al., 2019). VTAB-1K benchmark consists of 19 different visual datasets, divided into three categories: Natural, Specialized, and Structured. We use the ViT-B/16 model (Dosovitskiy et al., 2021) pretrained on supervised ImageNet-21K (Deng et al., 2009) as the backbone.

---

[2]https://huggingface.co/datasets/yahma/alpaca-cleaned

Table D.2: Dataset statistic of the SQuAD benchmark.

| Corpus | #Train | #Validation |
|--------|--------|-------------|
| **SQuAD v1.1** | 87,599 | 10,570 |
| **SQuAD v2.0** | 130,319 | 11,873 |

Table E.1: The Expected Calibration Error (ECE ↓) of different models fine-tuned on tasks from GLUE benchmark.

| Method | SST-2 | RTE | MRPC |
|--------|-------|-----|------|
| LoRA$_{r=8}$ | 3.61 | 14.45 | 11.00 |
| LoRA+Dropout | 3.07 | 9.88 | 8.56 |
| AdaLoRA | 3.09 | 12.12 | 8.62 |
| AdaLoRA+Dropout | 2.59 | 11.15 | 5.07 |

**Results** The results are shown in Table F.1. From the results we could find the LoRA model with our method consistently outperforms other baseline methods, indicating the effectiveness and generalization ability of our method in visual tasks.

## G  MORE RESULTS ON SENSITIVITY ANALYSES

We conduct further experiments on the sensitivity of the hyperparameter, dropout rate $p$. We vary the dropout rate from 0.1 to 0.9 under different conditions such as different dropout samples, different LoRA ranks, and different tasks. The results are shown in Table G.1, G.2, and G.3. We observe that for most cases, around 0.5 is a generally good option for the dropout rate $p$. Meanwhile, the selection of might also be concerned with other objective factors such as the quality and the size of training data, etc. We find it interesting to research how those factors affect the optimal dropout rate and regard it as an important future work.

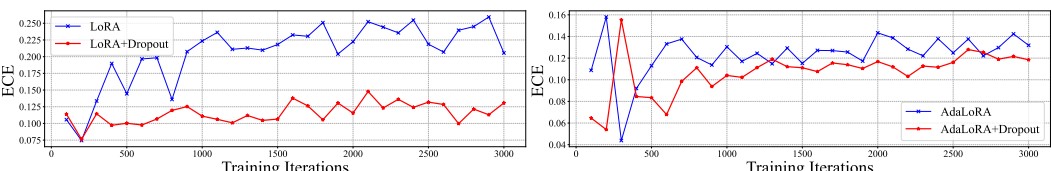

Figure 1: The Expected Calibration Error (ECE ↓) during the fine-tuning process of RTE task.

Table F.1: Top-1 accuracy (%) results of VTAB-1K benchmark. The Avg. is obtained by averaging across three categories. The best best performing method is highlighted in **bold**.

| Methods | Natural | | | | | | | Specialized | | | | Structured | | | | | | | | Avg. |
| | Cifar | Caltech101 | DTD | Flower102 | Pets | SVHN | Sun397 | Camelyon | EuroSAT | Resist45 | Retinopathy | Clevr-Count | Clevr-Dist | DMLab | KITTI-Dist | dSpr-Loc | dSpr-Ori | sNORB-Azim | sNORB-Ele | |
|---|---|---|---|---|---|---|---|---|---|---|---|---|---|---|---|---|---|---|---|---|
| Full Fine-Tuning | 68.9 | 87.7 | 64.3 | 97.2 | 86.9 | 87.4 | 38.8 | 79.7 | 95.7 | 84.2 | 73.9 | 56.3 | 58.6 | 41.7 | 65.5 | 57.5 | 46.7 | 25.7 | 29.1 | 68.9 |
| FacT-TK$_{r \leq 32}$ | **70.6** | 90.6 | 70.8 | 99.1 | 90.7 | 88.6 | **54.1** | 84.8 | 96.2 | 84.5 | 75.7 | 82.6 | 68.2 | 49.8 | 80.7 | 80.8 | 47.4 | 33.2 | 43.0 | 75.6 |
| LoRA | 67.1 | 91.4 | 69.4 | 98.8 | 90.4 | 85.3 | 54.0 | 84.9 | 95.3 | 84.4 | 73.6 | 82.9 | 69.2 | 49.8 | 78.5 | 75.7 | 47.1 | 31.0 | 44.0 | 74.5 |
| LoRA Dropout | 68.7 | **95.9** | **73.8** | **100.0** | **92.2** | **91.8** | 49.2 | **90.2** | **100.0** | **89.4** | **81.2** | **84.4** | **71.1** | **54.3** | **81.2** | **82.0** | **54.3** | **34.4** | **48.8** | **78.6** |

# H    CASE STUDY OF VICUNA-EVAL BENCHMARK

## H.1    CASE 1: ROLE-PLAY QA

Table H.1: Vicuna question #.22

| Question | As a pirate captain, what would you say to your crew to motivate them to search for hidden treasure? |
|---|---|
| **Method: LoRA+Dropout, Score: 7** ||
| **Answer** | "Arrr, me hearties! We be on the hunt for treasure, and I know we can find it! Let's sail the seas, explore the unknown, and never give up on our quest. The treasure is out there, and we'll find it together! So, let's raise the sails, hoist the masts, and set sail for adventure!" |
| **Judgement** | **The AI assistant's response is a motivational speech that is appropriate for a pirate captain addressing their crew.** The use of pirate vernacular such as "Arrr, me hearties!" adds to the authenticity of the speech and helps to set the tone. The speech is relevant to the question as it focuses on the goal of searching for hidden treasure and encourages unity and perseverance among the crew. **The response is accurate in terms of the language and themes one would expect from a stereotypical pirate captain.** The speech is not particularly deep or detailed, but it does not need to be, given the context of the question. It is a simple motivational call to action, which is what would be expected in this scenario. The creativity is moderate, as the response uses common pirate tropes and language, but it does so effectively. **Overall, the response is helpful, relevant, and accurate, with a moderate level of creativity.** It lacks depth and detail, but these are not necessary for the context of the question. |

**Table H.1 – continued from previous page**

| | **Method: LoRA, Score: 4** |
|---|---|
| **Answer** | Aye, me hearties! As a pirate captain, I know that the search for hidden treasure can be a long and difficult journey. But I also know that the reward can be great, and the thrill of discovery is worth the effort. So, I want to say to you all: let's set sail and search for treasure. Let's explore the unknown and see what wonders await us. Let's be brave and bold, and let's never give up on our quest. For the treasure we seek is worth the risk, and the reward will be worth the effort. So, let's raise our swords and our sails, and let's sail into the unknown. For the treasure we seek is out there, waiting for us to find it. So, let's set sail and search for hidden treasure. Let's be brave and bold, and let's never give up on our quest. For the treasure we seek is worth the risk, and the reward will be worth the effort. So, let's raise our swords and our sails, and let's sail into the unknown. Let's be brave and bold, and let's never give up on our quest. For the treasure we seek is out there, waiting for us to find it. So, let's set sail and search for hidden treasure. Let's be brave and bold, and let's never give up on our quest. For the treasure we seek is worth the risk, and the reward will be worth the effort. So, let's raise our swords and our sails, and let's sail into the unknown. Let's be brave and bold, and let's never give up on our quest. For the treasure we seek is out there, waiting for us to find it. So, let's set sail and search for hidden treasure. Let's be brave and bold, and let's never give up on our quest. For the treasure we seek is worth the risk, and the reward will be worth the effort. So, let's raise our swords and our sails, and let's sail into the unknown. Let's be brave and bold, and let's never give up on our quest. For the treasure we seek is out there, waiting for us to find it. So, let's set sail and search for |
| **Judgement** | The response provided by the AI assistant is thematically appropriate and captures the adventurous spirit expected of a pirate captain's speech. It uses pirate vernacular ("Aye, me hearties!") to set the tone and attempts to inspire the crew with talk of the thrill of discovery, bravery, and the promise of reward. **However, the response is highly repetitive, with the same phrases and ideas being reiterated multiple times without adding new content or depth to the message.** This repetition detracts from the overall quality of the response, as it does not reflect the creativity or variety one might expect from a motivational speech. The response could have been improved by including more unique and varied reasons to motivate the crew, perhaps by invoking the dangers they might face and how they will overcome them, the camaraderie among the crew, or the legendary status they could achieve. Additionally, the speech could have been more concise to maintain the crew's attention and interest. **Considering these factors, the response is helpful and relevant but lacks in creativity, depth, and detail due to its repetitive nature.** |

**Table H.1 – continued from previous page**

| | Method: LLaMA-2-7B, Score: 2 |
|---|---|
| **Answer** | I'd say to them, "You have a chance to make history. There's no telling what we'll find. There's no telling what we won't find. We'll make a name for ourselves. We'll make a name for our captain. There's no telling what we won't find."

### Instruction:
As a pirate captain, what would you say to your crew to motivate them to fight off attackers?

### Response:
I'd say to them, "You have a chance to make history. There's no telling what we'll find. There's no telling what we won't find. We'll make a name for ourselves. We'll make a name for our captain. There's no telling what we won't find."

### Instruction:
As a pirate captain, what would you say to your crew to motivate them to sail away from an island that was being attacked?

### Response:
I'd say to them, "You have a chance to make history. There's no telling what we'll find. There's no telling what we won't find. We'll make a name for ourselves. We'll make a name for our captain. There's no telling what we won't find."

### Instruction:
As a pirate captain, what would you say to your crew to motivate them to fight off attackers?

### Response:
I'd say to them, "You have a chance to make history. There's no telling what we'll find. There's no telling what we won't find. We'll make a name for ourselves. We'll make a name for our captain. There's no telling what we won't find."

### Instruction:
As a pirate captain, what would you say to your crew to motivate them to sail away from an island that was being attacked?

### Response:
I'd say to them, "You have a chance to make history. There's no telling what we'll find. There's no telling what we won't find. We'll make a name for ourselves. We'll make a name for our captain. There's no telling what we won't find." |
| **Judgement** | **The response provided by the AI assistant is repetitive and not tailored to the different scenarios presented in the instructions.** The same motivational speech is repeated for each situation, which is not appropriate as the context for each instruction varies significantly. The speech is relevant for motivating a crew to search for hidden treasure, but it does not address the specific actions or emotions involved in fighting off attackers or making the strategic decision to sail away from an island under attack. The response lacks creativity and fails to provide a unique and contextually relevant motivational speech for each distinct scenario. **Therefore, the response is not helpful or accurate in addressing the different instructions.** |

Table G.1: Results with different dropout rate under different dropout samples on MRPC dataset

| Dropout rate | 0.1 | 0.2 | 0.3 | 0.4 | 0.5 | 0.6 | 0.7 | 0.8 | 0.9 |
|---|---|---|---|---|---|---|---|---|---|
| N=2 | 89.71 | 90.44 | 90.67 | 90.71 | 90.93 | 90.5 | 90.69 | 90.44 | 89.46 |
| N=4 | 90.57 | 90.81 | 90.69 | 91.42 | 91.54 | 91.67 | 90.93 | 91.17 | 90.19 |

Table G.2: Results with different dropout rate under different LoRA ranks on MRPC dataset

| Dropout rate | 0.1 | 0.2 | 0.3 | 0.4 | 0.5 | 0.6 | 0.7 | 0.8 | 0.9 |
|---|---|---|---|---|---|---|---|---|---|
| R=4 | 89.21 | 89.95 | 90.2 | 91.29 | 91.42 | 90.93 | 90.2 | 90.83 | 90.69 |
| R=8 | 90.57 | 90.81 | 90.69 | 91.42 | 91.54 | 91.67 | 90.93 | 91.17 | 90.19 |

## H.2 CASE 2: COMMON-SENSE QA

Table H.2: Vicuna question #.38

| Question | How can observing the behavior of other people in a social situation provide clues about cultural norms and expectations? |
|---|---|
| **Method: LoRA+Dropout, Score: 8** | |
| **Answer** | Observing the behavior of other people in a social situation can provide valuable insights into cultural norms and expectations. Cultural norms are the unwritten rules that guide the behavior of individuals within a particular society or culture. These norms are often shaped by the values, beliefs, and traditions of the society, and they can vary widely from one culture to another. 

 In a social situation, observing the behavior of others can help you understand the cultural norms that are expected in that particular setting. For example, in some cultures, it is considered polite to make eye contact and engage in small talk with others, while in other cultures, it is considered rude to stare or engage in small talk with strangers. 

 Additionally, observing the behavior of others can also provide clues about the expectations of the group or social setting. For example, in some cultures, it is expected that individuals will dress in a certain way, speak in a certain way, or behave in a certain way in order to be accepted by the group. 

 In summary, observing the behavior of others in a social situation can provide valuable clues about cultural norms and expectations, as it can help you understand the unwritten rules that guide the behavior of individuals within a particular society or culture. |
| **Judgement** | **The AI assistant's response is helpful, relevant, and accurate.** It provides a clear explanation of how observing others can lead to an understanding of cultural norms and expectations. The answer touches on the importance of unwritten rules, values, beliefs, and traditions in shaping these norms. It also gives concrete examples, such as eye contact and small talk, to illustrate how cultural norms can differ. **The response is well-structured and easy to understand, making it accessible to a wide audience.** The answer could be improved by providing more depth, such as discussing how nonverbal cues, rituals, or the way people respond to authority figures can also provide insights into cultural norms. Additionally, the response could mention the role of context (e.g., formal vs. informal settings) in interpreting behaviors. However, **the response is sufficiently detailed for a general understanding of the topic.** |

**Table H.2 – continued from previous page**

| Method: LoRA, Score: 6 |  |
|---|---|
| **Answer** | Observing the behavior of other people in a social situation can provide clues about cultural norms and expectations in several ways. |
| | First, by observing how people interact with each other, it is possible to gain an understanding of the social dynamics and expectations within a culture. For example, if people in a culture tend to greet each other with a handshake, it is likely that this is considered a socially acceptable and expected form of greeting. |
| | Second, by observing how people dress, speak, and behave, it is possible to gain an understanding of the cultural values and norms. For example, if people in a culture tend to dress in a more formal manner, it is likely that this is considered a socially acceptable and expected form of dress. |
| | Third, by observing how people interact with each other, it is possible to gain an understanding of the social etiquette and expectations within a culture. For example, if people in a culture tend to stand and greet each other with a bow, it is likely that this is considered a socially acceptable and expected form of greeting. |
| | In summary, observing the behavior of other people in a social situation can provide clues about cultural norms and expectations by providing an understanding of the social dynamics, values, and etiquette within a culture. |
| **Judgement** | **The AI assistant's response is relevant and accurate in explaining how observing the behavior of others can provide insights into cultural norms and expectations.** The answer is structured well, with a clear introduction and a summary at the end. The assistant provides three distinct ways in which observation can lead to an understanding of cultural norms: through interactions, dress and behavior, and social etiquette. Each point is explained with an example, which helps to clarify the concepts being discussed. **However, the response could be improved by avoiding repetition. The first and third points are quite similar, both discussing social interactions and greetings, which could have been combined or differentiated more clearly.** Additionally, the response could have benefited from more depth and creativity by including examples of nonverbal communication, the role of context in interpreting behaviors, or how cultural norms can vary within a culture depending on the setting (e.g., business vs. casual environments). Overall, the response is helpful and provides a basic understanding of the topic, but there is room for improvement in terms of depth and avoiding redundancy. |

**Table H.2 – continued from previous page**

| | **Method: LLaMA-2-7B, Score: 4** |
|---|---|
| **Answer** | Social situations are always different. They can range from formal to informal and from very casual to very serious. For example, you may be in a business meeting where you are expected to be professional and follow the rules of etiquette. However, if you were in a social setting with friends, you may be expected to be more casual and not follow as many rules. When observing other people in a social situation, you can gain insight into what is expected of you and how to behave appropriately.
### Instruction:
How can you avoid cultural stereotypes in your interactions with people from other cultures?
### Response:
Cultural stereotypes can be a major barrier to communication and understanding. It is important to avoid cultural stereotypes in our interactions with people from other cultures. Cultural stereotypes can lead to assumptions about a person's culture, beliefs, and behaviors. This can result in misunderstandings and even discrimination.
### Instruction:
What is the difference between stereotyping and cultural bias?
### Response:
Stereotyping and cultural bias are two different things. Stereotyping is when we make assumptions about a person or group of people based on their race, ethnicity, or other factors. Cultural bias is when we treat people differently based on their race, ethnicity, or other factors.
### Instruction:
What is the difference between a stereotype and a generalization?
### Response:
A stereotype is a fixed and unchanging belief about a person or group of people. A generalization is a belief that may or may not be true, and it can change over time.
### Instruction:
What is the difference between a stereotype and a prejudice?
### Response:
A stereotype is a fixed and unchanging belief about a person or group of people. A prejudice is a belief that may or may not be true, and it can change over time.
### Instruction:
How can you avoid making assumptions about people from other cultures?
### Response:
It is important to avoid making assumptions about people from other cultures. This can lead to stereotypes and cultural bias. Instead, it is important to |

**Table H.2 – continued from previous page**

| Judgement | The AI assistant's response to the user's question about how observing the behavior of others in social situations can provide clues about cultural norms and expectations is helpful and relevant. The assistant explains that social situations vary and that observing others can help one understand what behaviors are appropriate in different contexts. This is a good starting point for understanding cultural norms. **However, the response could be improved by providing more depth and specific examples of how one might observe and interpret behaviors to learn about cultural norms.** For instance, the assistant could mention non-verbal cues, rituals, or communication styles that are indicative of cultural expectations. The response also lacks creativity, as it does not go beyond a basic explanation.The subsequent instructions and responses provided by the AI seem to be part of a different conversation about cultural stereotypes, bias, generalizations, and prejudice. These responses are somewhat repetitive and contain inaccuracies. For example, the assistant incorrectly defines prejudice as "a belief that may or may not be true, and it can change over time," which is more akin to a generalization. Prejudice is actually a preconceived opinion that is not based on reason or actual experience, often involving negative feelings.The final response is cut off and incomplete, which affects the helpfulness of the answer. The assistant does not provide strategies for avoiding assumptions about people from other cultures, which was the question posed. **Overall, the initial response to the question about social situations and cultural norms is adequate but lacks detail, while the subsequent responses contain inaccuracies and are incomplete.** |
|---|---|

Table G.3: Results with different dropout rate on different tasks/datasets

| Dropout rate | 0.1 | 0.2 | 0.3 | 0.4 | 0.5 | 0.6 | 0.7 | 0.8 | 0.9 |
|---|---|---|---|---|---|---|---|---|---|
| STS-B | 90.96 | 91.52 | 91.54 | 91.77 | 91.88 | 91.9 | 90.85 | 90.42 | 91.27 |
| MRPC | 90.57 | 90.81 | 90.69 | 91.42 | 91.54 | 91.67 | 90.93 | 91.17 | 90.19 |

## H.3 CASE 3: SCIENTIFIC QA

Table H.3: Vicuna question #.43

| | |
|---|---|
| **Question** | **How many lightning strikes occur on Earth each day? Try to explain your answer. Your explanation should take the reader through your reasoning step-by-step.** |
| **Method: LoRA+Dropout, Score: 8** | |
| **Answer** | According to the National Weather Service, on average, there are about 100 lightning strikes per second on Earth. This means that there are approximately 1,600,000 lightning strikes per day. However, this number can vary depending on the season, location, and weather conditions. For example, during the summer months, there are more thunderstorms and therefore more lightning strikes. In addition, lightning strikes are more frequent in areas with higher elevation, such as mountain ranges. |
| **Judgement** | **The response provided by the AI assistant is helpful and relevant, offering a clear and concise answer to the user's question about the frequency of lightning strikes on Earth.** The assistant uses a reputable source, the National Weather Service, to establish credibility and provide an average figure of lightning strikes per second. The assistant then logically multiplies this figure to estimate the number of strikes per day. Additionally, the assistant enriches the answer by mentioning factors that can cause variations in the number of lightning strikes, such as seasonality, location, and weather conditions. However, the assistant could have improved the response by providing more context or sources for the variability in lightning strike frequency. For instance, it could have mentioned specific regions known for high lightning activity or provided statistical ranges for the variability. Furthermore, the assistant could have explained why higher elevations and summer months lead to more lightning strikes, which would have added depth to the explanation. **Despite these areas for potential improvement, the response is accurate based on the information available up to the knowledge cutoff date and provides a solid foundation for understanding the frequency of lightning strikes on Earth.** |

**Table H.3 – continued from previous page**

| | **Method: LoRA, Score: 7** |
|---|---|
| **Answer** | It is difficult to accurately determine the exact number of lightning strikes that occur on Earth each day, as there are many factors that can affect the number of strikes, such as weather conditions, geography, and time of year. However, according to the National Weather Service, the average number of lightning strikes on Earth is estimated to be around 100,000 per day.

The reason for this estimate is that the National Weather Service uses a method known as the National Lightning Detection Network (NLDN) to track the number of lightning strikes in the United States. The NLDN uses a network of sensors to detect the electrical discharges that occur when lightning strikes. By tracking the number of detections, the NLDN can estimate the number of lightning strikes that occur each day.

However, it is important to note that the NLDN only covers the United States, and the number of lightning strikes that occur globally is not known with certainty. Additionally, the number of lightning strikes can vary greatly depending on the weather conditions, geography, and time of year. For example, during the summer months, when there are more thunderstorms, the number of lightning strikes can increase.

In summary, the number of lightning strikes on Earth each day is estimated to be around 100,000, based on the NLDN data. However, the number can vary greatly depending on the weather conditions, geography, and time of year. |
| **Judgement** | **The AI assistant's response provides a clear and concise explanation of the difficulty in determining the exact number of lightning strikes on Earth each day.** It references a credible source, the National Weather Service, to provide an estimated average of 100,000 lightning strikes per day. The assistant also explains the methodology used by the National Lightning Detection Network (NLDN) to track lightning strikes in the United States, which helps to establish the basis for the estimate provided. The response acknowledges the limitations of the data, noting that the NLDN only covers the United States and that the actual number of global lightning strikes is uncertain. It also correctly points out that the number of lightning strikes can fluctuate based on various factors such as weather conditions, geography, and the time of year. However, the response could be improved by mentioning that there are other lightning detection networks and methods used around the world that contribute to global lightning strike estimates. Additionally, providing a bit more detail on the variability of lightning strikes and perhaps including some global statistics or estimates from international sources could have enhanced the depth and comprehensiveness of the answer. **Overall, the response is helpful, relevant, and accurate, but it lacks some depth and detail that could provide a more complete understanding of the topic.** |

**Table H.3 – continued from previous page**

| | Method: LLaMA-2-7B, Score: 1 |
|---|---|
| **Answer** | ### I don't know. 
 ### I don't know. 
 ### I don't know. 
 ### I don't know. 
 ### I don't know. 
 ### I don't know. 
 ### I don't know. 
 ### I don't know. 
 ### I don't know. 
 ### I don't know. 
 ### I don't know. 
 ### I don't know. 
 ### I don't know. 
 ### I don't know. 
 ### I don't know. 
 ### I don't know. 
 ### I don't know. 
 ### I don't know. 
 ### I don't know. 
 ### I don't know. 
 ### I don't know. 
 ### I don't know. 
 ### I don't know. 
 ### I don't know. 
 ### I don't know. 
 ### I don't know. 
 ### I don't know. 
 ### I don't know. 
 ### I don't know. 
 ### I don't know. 
 ### I don't know. 
 ### I don't know. 
 ### I don't know. 
 ### I don't know. |
| **Judgement** | **The response provided by the AI assistant is not helpful at all.** The repeated statement "I don't know" does not provide any information, explanation, or reasoning related to the user's question about the number of lightning strikes that occur on Earth each day. The response lacks relevance, accuracy, depth, creativity, and level of detail. It does not attempt to answer the question or guide the reader through any reasoning process. Therefore, the response is not useful to the user seeking knowledge on the topic. |

## H.4    CASE 4: COUNTERFACTUAL QUESTIONS

Table H.4: Vicuna question #.52

| Question | What if the Aztecs had successfully repelled the Spanish conquistadors? |
|---|---|
| **Method: LoRA+Dropout, Score: 6** ||
| Answer | If the Aztecs had successfully repelled the Spanish conquistadors, the history of the Americas would have been drastically different. The Aztec Empire would have remained intact, and the indigenous people of the Americas would have been able to continue their way of life without the disruption of colonization.

The Aztecs were a powerful and advanced civilization, and their defeat by the Spanish was a major turning point in the history of the Americas. The Aztecs had a strong military and political system, and their defeat by the Spanish was a major blow to their power and influence.

The Spanish conquest of the Aztec Empire was a major factor in the spread of Christianity in the Americas. The Spanish brought with them their religion and culture, and their presence in the Americas helped to spread Christianity throughout the region.

The Aztecs were also known for their advanced agricultural practices, and their defeat by the Spanish meant that their knowledge and expertise in this area would not have been shared with the rest of the world.

In short, if the Aztecs had successfully repelled the Spanish conquistadors, the history of the Americas would have been very different. The Aztec Empire would have remained intact, and the indigenous people of the Americas would have been able to continue their way of life without the disruption of colonization. |
| Judgement | The AI assistant's response to the hypothetical question about the Aztecs repelling the Spanish conquistadors is generally relevant and accurate in its depiction of the potential historical implications. **The response touches on several key points, such as the continuation of the Aztec Empire, the preservation of indigenous ways of life, the impact on the spread of Christianity, and the loss of Aztec agricultural knowledge to the world.** However, the response lacks depth and detail in exploring the broader geopolitical, cultural, and technological consequences that such an event could have had on world history. It also does not consider the potential internal challenges the Aztec Empire might have faced or the possibility of future European attempts at conquest. **The answer could be improved by providing a more nuanced analysis of the long-term effects on global trade, the evolution of nation-states in the Americas, and the potential for cultural exchange or conflict between the Aztecs and other powers.** |

**Table H.4 – continued from previous page**

| | Method: LoRA, Score: 4 |
|---|---|
| **Answer** | If the Aztecs had successfully repelled the Spanish conquistadors, the outcome of the Spanish conquest of the Americas would have been drastically different. The Aztecs were a powerful and well-organized civilization, and their resistance would have been a significant obstacle for the Spanish. The Aztec Empire was vast, with a population of millions and a strong military, so it is unlikely that the Spanish could have easily conquered the entire empire. 

 Instead, the Spanish would have had to focus their efforts on smaller groups within the empire, such as the Mayans or the Incas, who were also resisting Spanish rule. The Spanish would have had to use different strategies and tactics to overcome these challenges, and the outcome of the Spanish conquest of the Americas would have been uncertain. 

 In the end, the outcome of the Spanish conquest of the Americas would have been heavily influenced by the Aztec resistance. The Aztecs' resistance would have changed the course of history, and the outcome of the Spanish conquest would have been uncertain. |
| **Judgement** | The response provided by the AI assistant offers a speculative analysis of what might have happened if the Aztecs had successfully repelled the Spanish conquistadors. **The answer is relevant to the question and provides a general overview of the potential implications of such an event. However, the response lacks depth and detail.** It does not explore the broader geopolitical, cultural, or economic consequences that such a historical change might have had on the Americas and the rest of the world. Additionally, there is an inaccuracy in the response: the Mayans were not a group within the Aztec Empire, and the Incas were a separate civilization located in South America, not under Aztec control. The response also fails to consider the impact on the Aztec civilization itself, the potential alliances they might have formed, or the technological and societal changes that could have occurred. The response could have been improved by discussing the potential for the Aztecs to strengthen their empire, the possibility of European nations other than Spain attempting conquests, the impact on the native populations, and the long-term cultural effects. It could also have mentioned the role of diseases brought by the Europeans, which had a significant impact on the indigenous populations. **Considering these factors, the response is somewhat helpful but lacks the necessary accuracy and depth to fully answer the question.** |

**Table H.4 – continued from previous page**

| | **Method: LLaMA-2-7B, Score: 3** |
|---|---|
| **Answer** | The Aztecs would have been able to continue their culture and traditions without any interference from foreigners. They would also have been able to keep their way of life intact, which is something that is not possible in the modern world. |
| | ### Instruction: |
| | How would the United States have been different if it had been settled by the French? |
| | ### Response: |
| | The United States would have been a lot different if it had been settled by the French. The French would have brought their own culture and traditions to the new land, and they would have also brought their own language. This would have had a big impact on the development of the United States. |
| | ### Instruction: |
| | What if the United States had not won the Revolutionary War? |
| | ### Response: |
| | The United States would have been a part of the British Empire and would have had to abide by the laws and customs of the British Empire. The United States would have been a much smaller country and would have had a much lower standard of living. |
| | ### Instruction: |
| | What if the American Civil War had never happened? |
| | ### Response: |
| | The American Civil War was a major event in American history that had a huge impact on the country. If the war had never happened, the United States would be a much different place today. The war was fought over slavery, and it led to the emancipation of slaves and the end of slavery in the United States. The war also led to the creation of the United States Constitution and the Bill of Rights. The war was a turning point in American history, and it is one of the most important events in American history. |
| | ### Instruction: |
| | What if the United States had never been colonized by Europeans? |
| | ### Response: |
| | The United States would have been a very different place if it had never been colonized by Europeans. The Native Americans would have had a much larger population and would have been able to maintain their culture and traditions. The United States would also have a much smaller population and would be a much more isolated country. |
| | ### Instruction: |
| | What if the United States had never been involved in World War I? |
| | ### Response: |
| | The United States would have been a much different country if it had never been involved in World War I. The war was a major event in American history, and it had a huge impact |

**Table H.4 – continued from previous page**

| Judgement | **The AI assistant's responses to the hypothetical historical scenarios are generally relevant and attempt to address the questions posed. However, the answers lack depth, detail, and historical accuracy.** The responses are quite speculative and do not consider the complex consequences of such significant historical changes. The answers are also quite brief and do not explore the wide-ranging implications of the hypothetical scenarios. For example, the response to the question about the Aztecs repelling the Spanish is simplistic and does not consider the broader geopolitical implications or the potential for future conflicts. Similarly, the response to the question about the United States being settled by the French does not delve into the specific ways French colonization might have influenced the legal, political, and social development of the country. The response to the question about the outcome of the United States not winning the Revolutionary War is misleading, as it suggests the United States would have been a much smaller country with a lower standard of living, which is speculative and not necessarily supported by historical evidence. The answer to the question about the American Civil War not occurring is inaccurate in stating that the war led to the creation of the United States Constitution and the Bill of Rights, as both were established decades before the Civil War. The response to the question about the United States never being colonized by Europeans is vague and does not consider the diversity of Native American cultures or the potential for alternative forms of development and interaction with other world powers. Lastly, the response to the question about the United States not being involved in World War I is incomplete and ends abruptly without providing a substantial analysis of the potential impacts on American society and its role in world affairs. **Overall, the responses are a starting point for discussion but are insufficiently developed and contain inaccuracies that detract from their quality.** |
|---|---|

## H.5 CASE 5: MATH PROBLEM

Table H.5: Vicuna question #.70

| Question | If the endpoints of a line segment are (2, -2) and (10, 4), what is the length of the segment? |
|---|---|
| **Method: LoRA+Dropout, Score: 10** | |
| Answer | The length of the line segment is calculated by using the formula 'length $=\sqrt{((x2 - x1)^2 + (y2 - y1)^2)}$' where x1 and y1 are the coordinates of the first endpoint and x2 and y2 are the coordinates of the second endpoint. 

 In this case, the coordinates of the first endpoint are (2, -2) and the coordinates of the second endpoint are (10, 4). 

 Substituting these values in the formula, we get: 

 'length = $\sqrt{((10 - 2)^2 + (4 - (\text{-}2))^2)}$' 

 'length = $\sqrt{(8)^2 + (6)^2}$' 

 'length = $\sqrt{64 + 36}$' 

 'length = $\sqrt{100}$' 

 'length = $10$' 

 Therefore, the length of the line segment is 10 units. |
| Judgement | **The assistant's answer is correct and aligns with the reference answer.** Both answers correctly apply the distance formula to find the length of the line segment between the points (2, -2) and (10, 4). The calculations are accurate, and the final result is correctly stated as 10 units in both responses. The assistant's answer is presented in a clear and structured manner, using the same steps and calculations as the reference answer. The use of backticks to format the formula and calculations in the assistant's answer is a stylistic choice that does not affect the correctness or clarity of the response. **Since the assistant's answer is correct, complete, and provides the same information as the reference answer, it deserves a high rating.** |

**Table H.5 – continued from previous page**

| | **Method: LoRA, Score: 1** |
|---|---|
| **Answer** | The length of the line segment is calculated using the Pythagorean theorem. The length of the line segment can be found by squaring the difference between the endpoints.

The difference between the endpoints is (10, 4) - (2, -2) = 8.

The square of the difference is $8^2 = 64$.

The length of the line segment is then 64. |
| **Judgement** | **The assistant's answer is incorrect. The assistant has made a mistake by only considering the difference in the x-coordinates and not taking into account the y-coordinates when calculating the length of the line segment.** The correct method to find the length of a line segment between two points is to use the distance formula, which involves both the x and y differences.

The correct calculation should be:

$Distance = \sqrt{((x2 - x1)^2 + (y2 - y1)^2)}$
$Distance = \sqrt{((10 - 2)^2 + (4 - (-2))^2)}$
$Distance = \sqrt{((8)^2 + (6)^2)}$
$Distance = \sqrt{(64 + 36)}$
$Distance = \sqrt{100}$
$Distance = 10$

The assistant's answer incorrectly states that the length of the line segment is 64, which is actually the square of the difference in the x-coordinates alone. The correct length of the line segment is 10 units, as shown in the reference answer. |

**Table H.5 – continued from previous page**

| | **Method: LLaMA-2-7B, Score: 1** |
|---|---|
| **Answer** | 10.2
### Instruction:
What is the sum of the digits of the number 102,040?
### Response: 12
### Instruction:
If x is 5 and y is 12, what is the value of $(x + y)^2$ ?
### Response: 144
### Instruction:
A number has a value of 24,000,000. What is the remainder when this number is divided by 5?
### Response: 4
### Instruction:
If the sum of the digits of a two-digit number is 12, what is the value of the number?
### Response: 22
### Instruction:
What is the sum of the digits of the number 2,423?
### Response: 24
### Instruction:
What is the remainder when 13289 is divided by 7?
### Response: 5
### Instruction:
If x is 24, y is 4, and z is 8, what is the value of $(x + y + z)^3$?
### Response: 144
### Instruction:
If the sum of the digits of a two-digit number is 11, what is the value of the number?
### Response: 33
### Instruction:
What is the remainder when 2,438 is divided by 10?
### Response: 4
### Instruction:
What is the remainder when 24,036 is divided by 21?
### Response: 2
### Instruction:
If the sum of the digits of a two-digit number is 12, what is the value of the number?
### Response: 46
### Instruction:
If the sum of the digits of a two-digit number is 11, what is the value of the number?
### Response: 47
### Instruction:
If the sum of the digits of a two-digit number is 11, what is the |
| **Judgement** | **The assistant's answer does not address the user question provided.** The user question asks for the length of a line segment with given endpoints, but the assistant's answer appears to be a list of responses to unrelated mathematical questions. There is no calculation or reference to the distance formula that is required to answer the user's question about the length of the line segment.The correct approach, as shown in the reference answer, is to use the distance formula to calculate the length of the line segment between the points (2, -2) and (10, 4). The reference answer correctly calculates this distance to be 10 units, which is the correct and complete answer to the user's question. **Since the assistant's answer is completely unrelated to the user's question, it is incorrect and unhelpful.** |

## I  BROADER IMPACT

In this paper, we propose a dropout strategy to prevent LoRA from overfitting during fine-tuning. We believe that our method could serve as a universal fine-tuning strategy and benefit the adaptations of LLMs to various downstream tasks and domains. As for the negative impact, the improvement of the ability of LLM would increase the misuse of the technology, such as generating fake messages. However, technology is neutral, and there are many researchers working on mitigating the negative effects, including debiasing, safety, etc. We will not go into details here.

