# OpenReview forum: "LoRA Dropout as a Sparsity Regularizer for Overfitting Reduction"
_ICLR.cc/2025/Conference — ICLR 2025 Conference Withdrawn Submission_

### Official Review · Reviewer_uknh · 2024-10-23

**Soundness:** 2
**Presentation:** 2
**Contribution:** 1
**Rating:** 3
**Confidence:** 5

**Summary:**

This paper proposes LoRA Dropout, a novel sparsity regularization technique that applies dropout to the low-rank matrices in LoRA-based fine-tuning methods to mitigate overfitting, offering both theoretical analysis and practical improvements across various NLP tasks.

**Strengths:**

1. I endorse the decision to exclude dropout from the rank dimension in the design of the method.
2. LoRA Dropout can help existing methods like LoRA achieve better performance.

**Weaknesses:**

1. Line 49-50, the sentence is written in bold. Does it relate to the motivation of this paper and LoRA Dropout? If so, what is the relationship between "high rank overfitting" and LoRA Dropout? Can LoRA Dropout alleviate the overfitting issues caused by a high-rank LoRA model?

2. Following up on the first question, I do not believe that AdaLoRA addresses the high-rank overfitting problem as stated. Instead, AdaLoRA aims to allocate different ranks to weights based on their importance, which is unrelated to the issue of high-rank overfitting. If the authors claim that AdaLoRA can resolve this issue, further explanation is required.

3. The motivation of this paper is unclear, and the introduction lacks logical coherence. In essence, the introduction's logic could be simplified to: high-rank causes overfitting, AdaLoRA mitigates this issue but is insufficient, and Dropout can address it, but without solid theoretical support. It appears that the fundamental problem this paper seeks to address is how to theoretically prove that Dropout can solve the high-rank overfitting issue. However, this is not the case. Could the authors clarify the true motivation behind this work?

4. I do not consider this manuscript to offer a theoretical analysis of the LoRA optimization process with dropout. First, the whole section 2 is quite similar to that in [1], the two main conclusions (Eqs. 5 and 7) differ from the formula 6 and 7 in [1] merely by a change in variables. The proofs in this paper is also heavily similar to those in [1]. Second, Eq. 1 in this paper can be transformed into Eq 1 in [1]. Therefore, they are essentially the same problem. Third, the conclusions drawn in Section 2 (lines 186-193) are, in fact, consistent with the analysis of the effects of sparsity in [1]. Dropout can be viewed as a specific form of sparsity, and therefore, the relationships proven in this section are already encompassed by the analysis in [1]. In conclusion, due to the similarity in proofs and the correlation between dropout and sparsity, I do not find Section 2 to be meaningful. The authors could have simply referenced the conclusions from [1] instead of reproducing the proofs from [1] and presenting them as their own contribution.

5.  While the method itself is not complex, it demands almost double the training time (Table C.2).

[1] On the effectiveness of parameter-efficient fine-tuning, AAAI 2023.

typos:
1. Line 13. faces
2. Line 212. bold delta W. Besides, the use of the symbol “delta W” in this paper is inconsistent.
3. Line 526. introduce
4. Table D.1 caption. dataset.
5. Table C.2 duplicated “on”
6. Table1 and 2 caption. Dropout
7. Line 436. between
8. Line 173, 909. Respectively

The writing of this paper gives the impression that it was completed rather hurriedly.

**Questions:**

1. Line 48-49, “these models typically tend to maintain a relatively high rank to ensure sufficient expressive power”. Indeed, as seen in many papers, with rank of 2,  LoRA series usually can achieve comparable performance to fully fine-tuning. Even fine-tuning LLAMA-7B, some methods can achieve comparable performance with rank=4.

2. What is the rank setting in NLU tasks? Additionally, how should the LoRA Dropout rate be configured? The original LoRA already includes a dropout rate—does it also play a role during training in this context?

3. Line 228, “which is equivalent to masking random columns..” LoRA Dropout indeed masks random columns and rows of Delta W. Why does masking rows improve performance? What would the outcome be if only rows were masked?

4. Line 240-243, “Additionally, performing dropout…sparsity in our framework.” I have a question: while I acknowledge that sparsity is important for fine-tuning, why is it necessary to increase the sparsity of delta W when designing LoRA Dropout? After all, more sparsity does not always equate to better performance.

---

> ### Comment · Reviewer_uknh · 2024-11-23
> **Reviewer Comment**
>
> Hi authors, I wanted to kindly follow up and inquire if the rebuttal has been completed. I look forward to hearing back from you at your earliest convenience.

---

> ### Author Response · Authors · 2024-11-25
> **Rebuttal to Reviewer uknh (1)**
>
> We sincerely appreciate your comments and suggestions. We made every effort to address all the concerns and revised our paper following your suggestions. In the following, we quote your comments and then give our detailed response point-by-point.
>
> > **W1: The relationship between "high-rank overfitting" and LoRA Dropout**
>
> The bold sentence is used to highlight the core problem of this paper——the overfitting issue. However, we do not mean by “overfitting only exists when the rank is high”, but instead, “it is hard to select a proper rank that could balance the expressiveness and overfitting risk”. Specifically, a relatively high rank of LoRA, which means more learnable parameters, indicates high expressive power, but as well makes the model prone to overfitting. Simply reducing the rank of LoRA (reducing number of learnable parameters) helps control overfitting but might lead to suboptimal performance. Thus, it is tricky to pick a proper rank that strikes a balance between overfitting and underfitting.
>
> On the other hand, Dropout methods (including our LoRA Dropout method) can address the overfitting issue without cutting the budget of parameter, and our theoretical results also support that Dropout can alleviate overfitting by introducing sparsity regularization instead of directly reducing parameters. Therefore, with the LoRA Dropout method, the LoRA model could maintain a relatively high rank (high expressive power) and meanwhile achieve a low risk of overfitting.
>
> >**W2: AdaLoRA and overfitting issue**
>
> In AdaLoRA, each LoRA matrix shares the same rank during initialization, and the ranks of LoRA matrices are automatically adjusted during training based on the importance scores. The AdaLoRA method shares the same insight of reducing the parameter budgets for controlling high-rank overfitting with the method that directly reduces LoRA ranks but in an automatic way. Here, we refer to the sentence from the original AdaLoRA paper, “Less importance ones (LoRA matrices) are pruned to have lower rank to **prevent overfitting** and save the computational budget.” However, the ability of AdaLoRA to alleviate overfitting is insufficient. The importance scores used in AdaLoRA are learned during training based on the training data, making the selected parameters less generalizable to unseen test data.
>
> >**W3: motivation and logic of the introduction section**
>
> The logic of the introduction resembles the logic that you summarized in W3, and we are sorry if we didn’t make it clear in the original paper. We have updated the introduction in the revised paper, and we’d like to summarize the current logic of the introduction here:
>
> LoRA models face the problem of overfitting. Simply reducing the rank of LoRA (reducing the number of learnable parameters) helps but might lead to suboptimal performance. AdaLoRA automatically prunes parameters to mitigate overfitting but is insufficient. Dropout can address the overfitting issue without cutting the budget of parameters, but the theoretical mechanism of its effectiveness on LoRA remains unclear. The motivation of our paper is to find out the theoretical evidence for Dropout to alleviate the overfitting issue on LoRA and to further propose a better dropout method on LoRA based on our theoretical framework.

---

> ### Author Response · Authors · 2024-11-25
> **Rebuttal to Reviewer uknh (2)**
>
> >**W4: theoretical contribution of the paper**
>
> We’d like to claim that there are intrinsic differences between our theory and [1] in multiple aspects.
>
> 1. The problem formulation in [1] does not actually fit LoRA tuning. [1] formulates LoRA tuning as a so-called “sparse” tuning algorithm. However, in practice, LoRA updates **all linear weights** with a low-rank **dense** residual module, which is actually **NOT** the case of [1]. The sparsity condition in [1] only holds when part of the linear matrices are tuned – yet still in a very rigid “sparsity” pattern, which is also not the mainstream practice currently.
> 2. Applying dropout mechanism to LoRA makes it a sparse tuning method, as we mentioned at the beginning of Section 2. However, the problem formulation in [1] is still NOT the case, because **dropout does NOT have a fixed sparsity pattern** as [1] proposed. In Eq.2-4 of [1], the sparsity mask M is fixed as long as the algorithm is determined, while dropout applies stochastic masks and does NOT fit this problem formulation. Hence, we refine the problem formulation with the expectation regularization term as shown in Eq.1-2.
> 3. As the problem formulation changes, the proofs are also produced in a different way due to the expectation term. Though proofs are still based on the PHS error bound theorem, we relax the conditions of the inequality for better generalization. First of all, we do not assume a sufficiently large sample size n as [1] did (See the condition of the inequality of $f_s(u)-f_s(v)$ in Lemma 1 [1], which actually may not lead to overfitting when data is sufficient). Moreover, taking suggestions from other reviewers, we no longer assume the Hessian matrix to be positive-semidefinite but instead constrain the regularization strength to be reasonably large, which is consistent with the practice of conducting hard dropout masking instead of soft regularization.
> To summarize, we believe that our theoretical results have different insights, more proper problem formulation, and more general applicable conditions than [1], which is of great significance to be presented in our paper.
>
> [1] On the effectiveness of parameter-efficient fine-tuning, AAAI 2023.
>
> >**W5: demanding more training time**
>
> We must admit that extra time overhead is a big limitation of our work, as we mentioned in the “Conclusions and Limitations” section. The additional training time makes it possible for our model to achieve better performance, and we are working to reduce the time overhead and achieve a better balance between performance and efficiency.
>
> >**Q1: LoRA models can achieve good performances with a small rank**
>
> Yes, sometimes the LoRA-based models do perform well with a small rank, but it does not indicate that it has achieved optimal performances. Smaller ranks have a lower risk of overfitting, while higher ranks have more expressive power. Therefore, it is tricky to pick a proper rank. Hopefully, with the proposed LoRA Dropout method, the LoRA model could maintain a relatively high rank (high expressive power) and meanwhile have a low risk of overfitting.
>
> > **Q2: The setting of rank and LoRA Dropout**
>
> We use rank=8 in the NLU tasks. And for the dropout rate, empirically we find values around 0.5 will all lead to a fine result for most cases, and the dropout rate is set to 0.5 in all our experiments. Furthermore, we do not use the original dropout in LoRA when LoRA Dropout is applied, and we have compared the performances between LoRA Dropout and original dropout.

---

> ### Author Response · Authors · 2024-11-25
> **Rebuttal to Reviewer uknh (3)**
>
> >**Q3: Results of masking rows in dropout**
>
> Here we provide the results on NLU tasks that only rows were masked.
>
> |                                    | MNLI  | SST2  | COLA  | QQP   | QNLI  | RTE   | MRPC  | STSB  | Avg   |
> |------------------------------------|-------|-------|-------|-------|-------|-------|-------|-------|-------|
> | dropout on columns                 | 90.07 | 94.26 | 70.87 | 91.66 | 94.44 | 86.64 | 90.20 | 91.60 | 88.72 |
> | dropout on rows                    | 90.14 | 94.61 | 71.17 | 91.6  | 94.22 | 85.92 | 90.20 | 91.52 | 88.67 |
> | LoRA Dropout (on columns and rows) | 90.85 | 95.87 | 71.32 | 92.22 | 94.56 | 88.09 | 91.42 | 92.00 | 89.54 |
>
> From the results, we could find that only masking rows does not lead to obvious improvements in performance. The reason for better performances of LoRA Dropout is that by simultaneously masking random columns and rows, LoRA Dropout could provide **much more diverse sparsity patterns** across the parameter space than only masking columns or rows.
>
> As shown in Eq.(3) from Section 2.1, effectively applying the dropout mechanism in LoRA training requires sampling sufficiently diverse sparsity patterns (d from the Bernoulli distribution) for better estimation of the expectation term. An ideal way is generating 0-1 masks for each element in the delta parameters. However, this method is unrealistic since generating such sparsity masks will lead to great GPU memory overhead (i.e., as large as the original weight matrices).
>
> In LoRA Dropout, by masking both rows and columns, generated sparsity patterns become more diverse and comprehensive than simply masking columns or rows, while GPU memory overhead is not significantly increased. This means our dropout strategy offers a better estimation of the sparsity regularizer in Eq.(3).
>
> > **Q4:  Why increasing the sparsity leads to better performance.**
>
> Sorry for the confusion. In the lines mention in Q4, we try to discuss the unnecessity of performing dropout on the rank dimensions, not the necessity of increasing the sparsity of parameters. Suppose we conduct dropout on the rank dimensions of matrices A and B, the production matrix $\Delta W$ will not be a sparse matrix, leading to an invalid sparsity regularizer.
>
> Moreover, as we explained in Q3, the reason we conduct dropout on both columns and rows of $\Delta W$ is not to increase the sparsity but to get more diverse sparsity patterns. And we do agree that more sparsity does not always equate to better performance. As shown in Eq.(7) of our paper, only a proper sparsity ratio (dropout rate) will lead to a good balance between overfitting and underfitting.

---

> ### Author Response · Authors · 2024-12-01
> **A kind reminder as the deadline is approaching**
>
> Dear Reviewer uknh,
>
> I hope this message finds you well.
>
> I am writing to kindly remind you that the deadline for the discussion phase regarding our paper is drawing close. We greatly appreciate the time and effort you have already dedicated to reviewing our work, and we have made point-by-point responses to your constructive suggestions and questions. However, it appears that we have not yet received your response to our rebuttal, and we are wondering if we have addressed your concerns so far. Your insights and feedback are invaluable to us, and we believe they have significantly enhanced the quality of our manuscript.
>
> After carefully addressing the comments and corresponding revisions, we kindly request your consideration of a more positive evaluation of our work if deemed appropriate. If you require any additional information or clarification from our end, please do not hesitate to reach out to us. We are more than willing to assist you in any way possible to ensure a thorough and constructive review process.
>
> Thank you once again for your time and consideration. We are looking forward to hearing from you soon.
>
> Warm Regards

---

> > ### Comment · Reviewer_uknh · 2024-12-01
> >
> > Dear authors,
> >
> > I sincerely apologize for my delayed response. To be honest, I was under the impression that I had already replied…
> >
> > Getting straight to the point, in my previous review, I mentioned that the theoretical proofs in this paper are almost identical to those in [1]. In response to my concern, the authors outlined three points in W4, claiming differences between their work and the theoretical framework of [1]. However, I believe the authors’ understanding of [1] in W4 is somewhat flawed, and these points contain certain inaccuracies.
> >
> > 1. If the authors had carefully reviewed [1], they would realize that the concept of sparsity in [1] is entirely different from what they described in their response. In [1], the original model is treated as a combination of the original model and the LoRA branch. Consequently, during the fine-tuning phase, what is effectively adjusted are only a very small number of weights in this equivalent model, which constitutes the sparsity. Therefore, this sparsity is independent of whether the updates in LoRA are dense or sparse. Moreover, it does not refer to the statement: “The sparsity condition in [1] only holds when part of the linear matrices are tuned.”
> >
> >
> > 2. In the section “Parameter Efficient Fine-tuning as Sparse Fine-tuned Model” of [1], it is evident that [1] does not impose a strict requirement for M to be fixed. M is merely treated as a given parameter for the convenience of subsequent analysis. Moreover, M can certainly vary; the original problem naturally transforms into a bi-level optimization problem, and this does not affect the validity of treating M as fixed in the context of the current optimization process. Furthermore, the $d$ in the authors’ manuscript can similarly be viewed as fixed.
> >
> > 3. Given the above two points, I believe the authors’ modifications to the theoretical framework of [1] are minimal. I remain convinced that the theoretical proofs in the authors’ work are almost equivalent to those in [1].
> >
> > [1] On the effectiveness of parameter-efficient fine-tuning, AAAI 2023.

---

### Official Review · Reviewer_i71V · 2024-10-24

**Soundness:** 3
**Presentation:** 3
**Contribution:** 2
**Rating:** 6
**Confidence:** 3

**Summary:**

The paper builds upon Low-Rank Adaption (LoRa) by introducing a dropout-based sparsity regulator to mitigate overfitting. The authors provide empirical evidence for signs of overfitting with LoRa and derive a generalisation error bound under sparsity regulation Their method drops rows and columns from both tuneable low-rank parameter matrices and maintains efficiency with an ensemble of losses with different dropout instances. Further, model inference is improved by an ensemble strategy compressing the error bound. Finally, all claims are empirically evaluated in Natural Language Understanding, Question Answering, and Instruction Tuning.

**Strengths:**

* The idea is empirically and theoretically well motivated.
* The improvements of DeBERTaV3 and LLaMA2-7B on GLUE, SQuAD, MMLU, and Vicuna-Eval benchmarks are statistically significant.
* The presentation is clear and easy to follow.

**Weaknesses:**

* The evaluation is focused on smaller and nowadays partially outdated models. The paper would benefir from more recent models.
* The ablation is missing test-statistics and standard deviations and is conducted on different, smaller datasets.
* The runtime increases drastically (For example, 18min -> 60min on CoLa).
* It is not clear, how model size further impacts runtime and efficiency as more parameters are involved. As inference time is very crucial it would be beneficial to explore this in more detail and give reccomendations.

**Questions:**

* Why did you decide to only DeBERTaV3 and LLaMA2-7B (for just one task) to validate your claims?
* Why did you conduct the ablation analysis only on MRPC and CoLA?
* How would the runtime look like for larger, state-of-the-art language models? Do you expect a similar increase in runtime and is this feasible in practive?

---

> ### Author Response · Authors · 2024-11-26
> **Rebuttal to Reviewer i71V**
>
> We sincerely appreciate your detailed comments and constructive suggestions. We made every effort to address all the concerns. In the following, we quote your comments and then give our detailed response point-by-point.
>
> >**W1/Q1: The reason of using DeBERTaV3 and LLaMA2-7B for evaluation**
>
> We choose the above two models following the common practice in previous works.
> DeBERTaV3 and LLaMA2-7B are two most commonly adopted pre-train models for evaluating the effectiveness of PEFT methods. DeBERTa (or its older BERT-based twin Roberta) is a classical model that has been adopted in the experiments of many PEFT papers[1,2,3,4], and LLaMA2-7B is used to evaluate the performance in larger and more advanced models, and is also widely adopted [2,3,4].
>
> Still, we find your suggestion of conducting experiments on more recent models quite instructive. We are currently running experiments on LLaMA3-8B, and will update the results as soon as they come out.
>
> [1] AdaLoRA: Adaptive Budget Allocation for Parameter-Efficient Fine-Tuning. ICLR 2023
>
> [2] VeLoRA: Memory Efficient Training using Rank-1 Sub-Token Projections. NeurIPS 2024
>
> [3] LoRA+: Efficient Low Rank Adaptation of Large Models. ICML 2024
>
> [4] Parameter Efficient Quasi-Orthogonal Fine-Tuning via Givens Rotation. ICML 2024
>
> >**W2/Q2: About the results in ablation studies.**
>
> In fact, the ablations are conducted on all 8 tasks of the GLUE benchmark. Here, we provide the results along with the standard deviations.
>
> |             | MNLI        | SST2        | COLA        | QQP         | QNLI        | RTE         | MRPC        | STSB         |
> |-------------|-------------|-------------|-------------|-------------|-------------|-------------|-------------|--------------|
> | NoDrop      | 90.65(0.07) | 94.95(0.33) | 69.82(0.53) | 91.99(0.04) | 93.87(0.24) | 85.20(0.51) | 89.95(0.39) | 91.60(0.10)  |
> | Drop$_{train}$ | 90.64(0.05) | 95.53(0.23) | 70.69(0.40) | 91.81(0.07) | 94.55(0.16) | 87.73(0.37) | 90.69(0.38) | 91.67(0.23)  |
> | Drop$_{test}$  | 88.45(0.10) | 90.60(0.37) | 68.36(0.82) | 87.89(0.08) | 90.59(0.52) | 84.11(0.68) | 85.29(0.67) | 89.43(0.39)  |
> | Drop        | 90.85(0.05) | 95.87(0.16) | 71.32(0.59) | 92.22(0.05) | 94.56(0.21) | 88.09(0.77) | 91.42(0.35) | 92.00(0.11)  |
>
>  We can find the results are still consistent on other tasks besides MRPC and CoLA.
>
> >**W3/W4/Q3: About the runtime of LoRA Dropout in larger models and its feasibility in practice**
>
> The runtime (both training and inference time) is related to the number of sampled dropout instances. Basically, the time is linear with $n_{sample}$, since the LoRA Dropout aggregates outputs from $n_{sample}$ dropout instances. Therefore, for larger models that require more time for a single forward pass, LoRA Dropout does further increase the training and inference time.
>
> However, the computational overhead would not make LoRA Dropout unusable in practice.
> First of all, one can adjust the number of samples to conduct a trade-off between overfitting control and training/inference time cost on their demand.
>
> Moreover, for training, the overfitting problem will practically become severe only when the data amount is limited. Under this scenario, the training time will not be a great problem (training time is short for a limited data scale) in comparison with an unacceptable overfitted performance.
>
> For inference, the test time ensembling could be replaced with a re-scaling method at the cost of a slight decrease in performance. In the re-scaling method, dropout is not conducted during testing, and the outputs are re-scaled based on the dropout rate to adapt to the train-test difference. Without ensembling, the re-scaling method requires no additional cost of inference time.
> As for the performance, please refer to the results of Drop$_{train}$ from the table in the above response, which indicate the performance with the re-scaling method (conducting dropout during training and re-scaling during testing).

---

### Official Review · Reviewer_RPQn · 2024-10-28

**Soundness:** 3
**Presentation:** 2
**Contribution:** 3
**Rating:** 5
**Confidence:** 4

**Summary:**

The paper proposes a dropout method for LORA fine-tuning which amounts to using dropout on the implicit hidden layer that sits in the middle of the low rank adapters. This is supported by questionable theoretical arguments and solid empirical evaluation.

**Strengths:**

* The idea is simple and makes a lot of sense.

* The empirical evaluation is comprehensive and convincing. In particular, I like figure 5 because it shows that the authors have identified a key question and are bringing a satisfactory answer.

**Weaknesses:**

The theoretical part (section 2) is weak and confuses the message of the paper.

* First, equation (1) is nonsensical. This constraint trivially implies that \delta\theta=0.  This can be fixed by using an inequality constraint instead (i.e. the constraint is less than an arbitrary constant C whose choice is related to the choice of lambda in the dual formulation).

* Second, theorem 2.4 does not rely on the presence of dropout, and therefore does not say much about dropout at all. The actual argument (lines 189 to 193) could have been made without this theorem. Therefore the theoretical development of section 2 seems irrelevant to the question at hand and distracts rather than inform the reader.

* Finally the dropout scheme discussed in section 2 (equation 3) is different from the scheme proposed in section 3.2 and used in the experiments (as explained by the authors in section 3.2). This further weakens the theory of section 2.

The paper would be as readable and informative if section 2 were replaced by developing the argument of lines 189 to 193 in the context of the actual dropout algorithm (the one of section 3.2). This would make the paper lighter but better overall.  Alternatively it may be possible to make theorem 2.4 dependent on the presence of dropout, preferably in the form used in the experiments. That would save the theoretical section.

**Questions:**

* Can the statement of theorem 2.4 depend on the presence of dropout (preferably in the form used in the experiments) and show that dropout contributes to reducing the generalization gap?

* What is the practical cost of the test time ensembling (in the experimental results)?

* In figure 4, what is the rank used for LORA-dropout?  This is important to grasp the meaning of the dropout rate.

---

> ### Comment · Reviewer_RPQn · 2024-11-25
>
> I noticed a revision of the paper that removes the nonsensical equation (1). This was already factored in my score.
>
> The revision also makes some changes in the theorem, without really addressing my objections, which are that (1) the theorem does not explicitly discusses dropout, (2) the actual argument could have been made without the theorem, and (3) and it would have applied to the actual algorithm instead of the impractical one discussed in the theoretical section.
>
> I realize that there is much pressure to have a theoretical section. However I also believe that a theoretical section does not help when it so loosely connected to the actual algorithms. On the other hand, the empirical results are convincing enough for me. This has not changed. Therefore I maintain my score.

---

> ### Author Response · Authors · 2024-11-26
> **Rebuttal to Reviewer RPQn**
>
> We sincerely appreciate your suggestions and positive comments on our empirical results. We made every effort to address all the concerns and have uploaded a revised version of the paper following your suggestions. In the following, we quote your comments and then give our detailed responses point-by-point.
>
> >**W1: Equation (1) is nonsensical.**
>
> Thank you for your suggestion, and we change the problem formulation to $\min_{\Delta  \theta} \mathcal{L}(\theta^0 +  \Delta \theta), \ \mathrm{s.t.} \ \mathbb{E}_{d \sim \mathrm{Bern}(p)}  ||d \odot \Delta \theta||_2^2 \leq c$, where $c$ is a constant. However, incorporating c does not influence the duality form of this optimization problem, which is mainly discussed in our theoretical analyses, as $c$ can be eliminated as a constant. Hence, the correctness of our theoretical results is still ensured. We have made the corresponding modifications in our revised manuscript.
>
> >**W2/Q1: Theorem 2.4 does not rely on the presence of dropout**
>
> In fact, Theorem 2.4 does have strong connections with dropout. **The dropout rate $p$ appears in the rightmost term of Eq.(7) in theorem 2.4**, and it is the only hyper-parameter that does not depend on the problem formulation and model architecture. When the dropout rate p increases, the value of the rightmost term of Eq.(7) decreases, indicating the generalization gap is reduced. By varying the dropout rate p from 1 to 0, we could derive the argument of lines 189 to 193 (lines 193 to 197 in the revised paper). Though the argument seems to be common in practice, our Theorem does offer solid theoretical evidence for it.
>
> >**W3: Mismatch of dropout scheme in section 2 and section 3**
>
> We admit that the current practice of LoRA Dropout is not an exact match to the regularized problem in Eq.(3), yet the ideal solution is infeasible in practice and our solution is actually a more proper way to approximate the strict Bernoulli mask.
>
> Specifically, in section 2, we discussed the regularized problem with dropout instances sampled from a Bernoulli distribution. An ideal way to achieve this scheme is generating 0-1 masks for each element in the updated parameters. However, this method is unrealistic since such dropout masks will be as large as model parameters, which is quite huge in current LLMs. Generating such dropout masks will lead to great GPU memory overhead.
>
> **Therefore, we hope to strike a trade-off between the theoretical scheme and practical feasibility.**
> In section 3, we propose LoRA Dropout, trying to generate as many sparsity patterns (dropout patterns) as we can without significantly increasing GPU memory overhead. By masking both rows and columns, LoRA dropout provides more diverse sparsity patterns and offers a better estimation of the sparsity regularizer in Eq.(3) than the original dropout mechanism, where only rows are dropped out. Therefore, our practical contribution lies in **a more memory-efficient and expressive approximation of the ideal dropout scheme**.
>
> >**Q2: What is the practical cost of the test time ensembling**
>
> The cost of test time ensembling is related to the number of sampled dropout instances. Basically, the time is linear with n_sample, since the results should be aggregated from n_sample models. One can adjust the number of samples to conduct a trade-off between performance and inference time cost on their demand.
>
> Moreover, the test time ensembling could be replaced with a similar re-scaling method as in the original dropout method. In the re-scaling method, dropout is not conducted during testing and the outputs are re-scaled based on the dropout rate to adapt to the train-test difference.
> Compared with the test time ensembling method, this method requires no additional cost of time, but will lead to a decrease of performance. Here we provide the results using the re-scaling method compared with test time ensembling.
>
> |                    | MNLI   | SST2   | COLA   | QQP    | QNLI   | RTE    | MRPC   | STSB    |
> |--------------------|--------|--------|--------|--------|--------|--------|--------|---------|
> | test-time ensemble | 90.85  | 95.87  | 71.32  | 92.22  | 94.56  | 88.09  | 91.42  | 92.00   |
> | re-scaling         | 90.64  | 95.53  | 70.69  | 91.81  | 94.55  | 87.73  | 90.69  | 91.67   |
>
> >**Q3: What is the rank used for LORA Dropout in Figure 4**
>
> The rank is set to 8 for the experiments shown in Figure 4. We have also conducted experiments of varying the dropout rate on different ranks like 4. Here, we provided the results on r=4 and r=8, respectively.
>
> | dropout rate | 0.1   | 0.2   | 0.3   | 0.4   | 0.5   | 0.6   | 0.7   | 0.8   | 0.9    |
> |--------------|-------|-------|-------|-------|-------|-------|-------|-------|--------|
> | rank=4       | 90.69 | 90.83 | 90.38 | 90.93 | 91.42 | 91.29 | 90.2  | 89.95 | 89.21  |
> | rank=8       | 90.57 | 90.81 | 90.69 | 91.42 | 91.54 | 91.67 | 90.93 | 91.17 | 90.19  |

---

> > ### Comment · Reviewer_RPQn · 2024-11-26
> >
> > Sorry for not expressing my concern well enough.
> >
> > Given that $d$ is drawn independently from $\Delta\theta$, the constraint expression of equation (1) can be rewritten
> >
> > $ \mathbb{E}_{d \sim \mathrm{Bern}(p)} ||d \odot \Delta \theta ||^2 $
> >
> > $ = \sum_i \mathbb{E}_{d \sim \mathrm{Bern}(p)} [d_i] ~ \Delta\theta_i^2 = \sum_i  p  \Delta\theta_i^2 = p || \Delta \theta||^2 $
> >
> > So this resembles L2 regularization with parameter $p\lambda$ instead of $\lambda$.
> > In the theorem, $\lambda$ and $p$ always appear together in the product $\lambda p$.
> > This is interesting, but, without the structured nature of dropping out states instead of weights, this not really about dropout.
> > Fortunately the actual algorithm is much more interesting.

---

> ### Author Response · Authors · 2024-11-28
> **Rebuttal to Reviewer RPQn #2**
>
> Thank you for your careful reading and further questions.
>
> Firstly, we have also made the similar derivation of Eq.(1) in Proposition A.2 in the Appendix, and we are pleased to see your careful deduction reaches the same conclusion as we did. This also explains why $\lambda$ and $p$ always appear together in our theoretical results, as $\lambda p$ is the l2-regularization coefficient.
>
> Secondly, as for what our theoretical results actually tackle, we'd like to claim that we are not studying the specific vanilla dropout scheme that randomly masks the neurons of hidden representations. Instead, we are studying a more general regularization strategy that drops out the parameter space, such as DropConnect [1] as well as vanilla dropout (which can be regarded as dropping the rows of parameter space). Therefore, our theoretical results apply to a more universal family of "dropout" regularization strategies. Furthermore, to obtain a better estimation of the Bernoulli mask of the entire parameter space as introduced in our theorems, we design the algorithm in Section 3, which enjoys a more diverse sparsity pattern than vanilla dropout without introducing large memory overheads.
>
> Thank you again for your suggestions, and we sincerely hope our explanations could resolve your doubts.
>
> [1] Wan, L., Zeiler, M., et al. Regularization of Neural Networks using DropConnect. ICML 2013

---

> ### Comment · Reviewer_RPQn · 2024-12-01
>
> Here is how I interpret your last reply:  The theorem indeed has very little to do with dropout because the so-called "dropout" parameter $p$ acts just as a multiplier on the L2 hyper-parameter $\lambda".  And you knew it since you even wrote this important piece of information it in the appendix...
>
> Even if we care about explicit or implicit L2 regularization instead of dropout, the theorem does not tell you much we did not know, since the $p\lambda$ controls the effective size of the set of weights (fixed pretraining weights and adjustable lora weights).
>
> Therefore it is my opinion that the theoretical argument lowers the quality of the paper rather than enhances it.
>
> Note that your actual dropout idea is valuable.  But it would be much more scientific to say that you do not have a precise theoretical argument but have a good hand-waving explanation (the one in lines 189 to 193 of the original paper). And it might be all that matters since we still have to deal with all the dark corners of deep network training.

---

### Official Review · Reviewer_jLmX · 2024-10-30

**Soundness:** 1
**Presentation:** 1
**Contribution:** 2
**Rating:** 3
**Confidence:** 4

**Summary:**

This paper investigates effectiveness of dropout for parameter-efficient fine-tuning (PEFT) methods, specifically LoRA. This topic has been investigated before. The authors first present an analysis about the effect of dropout (with Bernoulli noises) to the loss and generalization error of the learning algorithm. The authors show that a suitable use of dropout rate can reduce generalization error, which is similar with the existing understanding about dropout in ML. The authors then propose a new way (called LoRA dropout) to enclose dropout in LoRA (and LoRA-based variants). Different with prior methods, LoRA dropout applies dropout for the rows or columns of the tunable low-rank matrices. Furthermore, the authors propose an ensemble method for the inference phase, borrowing the idea of MC dropout. Such an ensemble can boost performance of LoRA dropout. Finally, the authors did an extensive experiment to evaluate performance of their method, in comparison with related baselines. The empirical results suggest that their new method can often perform better than the baselines on different benchmarks and tasks. This is really encouraging.

**Strengths:**

**Originality:**
This paper proposes the use of dropout for the rows or columns of the tunable low-rank matrices in LoRA, which is practical and overcomes the memory limitation of some prior dropout methods. Furthermore, an ensemble method is proposed for the inference phase to further boost quality of PEFT.

**Quality:**
The experimental results suggest that the proposed method can work well.

**Clarity:**
The writing is quite easy to follow.

**Significance:**
The proposed method seems to work well, which potentially is significant to many practical applications.

**Weaknesses:**

Despite some encouraging empirical results, many concerns arise from the current presentation. Those concerns are mostly for their theoretical analysis and writing. Each will be discussed below.

**For the theory:**

- The authors use algorithmic stability to analyze generalization error of dropout training. They analyze a regularized loss, which uses Bernoulli dropout in the regularization term. I am not sure that LoRA dropout training can be easily formulated as minimizing this simple regularized loss. Hence, some of their theoretical understandings may not apply to LoRA dropout.

- The main theoretical understandings are mostly originated from Proposition 2.2 which analyzes point-wise hypothesis stablity of a learning algorithm. However, there are many concerns about correctness and meaning of this proposition:

    - Closeness between $\mathbf{\theta}_L (\mathbf{S}^i)$ and $\mathbf{\theta}_L (\mathbf{S})$ is *not defined* clearly. Such quantity is very important to derive their result, and should play a significant part in the stability constant. The authors ignored this part completely in their proof, causing a big question about the goodness/meaning of their result.
    - The assumption about $\eta$-Lipschitzness of the loss is very important for their analysis and result. Such Lipschitzness seems to be w.r.t the model parameters. However, the authors did not clearly specify it. More importantly, the magnitude of $\eta$ should affect significantly to the bound on generalization error in Theorem 2.4. Nonetheless, the authors did not discuss how large $\eta$ is in practice. For big models, e.g., Llama2, $\eta$ might be very large. If so, their error bound in Theorem 2.4 will be vacuous or meaningless.
    - The assumptions about $\eta$-Lipschitzness and closeness should be translated to the training algorithm. Those assumptions are really strong, and may not fit well with practice.

- Proposition 3.1 claims that their ensemble classifier should have smaller error than the average error of the whole classifier family. However, the statement and the proof of this proposition have a big mismatch. Their proof in Appendix A.2 assumes that the loss is convex w.r.t. the model parameters. Such an assumption is entirely different from that in Proposition 3.1. More importantly, such an assumption is **unrealistic** for deep neural networks, since it is well-known that the training loss for DNNs is often nonconvex. Therefore, their interpretation from this proposition seems not to be well-supported.

**For the writing:**

- Some notations are used without definition, e.g., $\mathbf{\theta}_L$
- Some important concepts or assumptions are not defined explicitly, e.g. "closeness" and $\eta$-Lipschitzness.
- Some mistakes appear in Eq. (1) and (3). The authors misused the function and problem definitions.
- The loss function in Lemma 2.3 seems wrong.
- The empirical and expected losses in Lemma 2.3 and Theorem 2.4 seem wrong. Should they represent the loss for a learning algorithm?

**Questions:**

- Can the authors specify how significantly can "closeness" affect the stability constant in Proposition 2.2? Is closeness small for LoRA Dropout?
- How large is $\eta$? Is this small for your pretrained models?

---

> ### Author Response · Authors · 2024-11-26
> **Rebuttal to Reviewer jLmX**
>
> We sincerely appreciate your detailed comments and constructive suggestions. We made every effort to address all the concerns and have uploaded a revised version of the paper following your suggestions. In the following, we quote your comments and then give our detailed response point-by-point.
>
> >**W1: Differences between theoretical analyses and LoRA Dropout practice**
>
> We admit that the current practice of LoRA Dropout is not an exact match to the regularized problem in Eq.(3), yet the ideal solution is infeasible in practice and our solution is actually a more proper way to approximate the strict Bernoulli mask.
> Specifically, in section 2, we discussed the regularized problem with dropout instances sampled from a Bernoulli distribution. An ideal way to achieve this scheme is generating 0-1 masks for each element in the updated parameters. However, this method is unrealistic since such dropout masks will be as large as model parameters, which is quite huge in current LLMs. Generating such dropout masks will lead to great GPU memory overhead.
>
> **Therefore, we hope to strike a trade-off between the theoretical scheme and practical feasibility.**
> In section 3, we propose LoRA Dropout, trying to generate as many sparsity patterns (dropout patterns) as we can without significantly increasing GPU memory overhead. By masking both rows and columns, LoRA dropout provides more diverse sparsity patterns and offers a better estimation of the sparsity regularizer in Eq.(3) than the original dropout mechanism where only rows are dropped out. Therefore, our practical contribution lies in a more memory-efficient and expressive approximation of the ideal dropout scheme.
>
> >**W2.1/Q1: The “Closeness” between $\theta _L (S^i)$ and $\theta _L (S)$ needs to be defined clearly**
>
> Thank you for pointing this out. We have re-examined our proof of Proposition 2.2 and made it clear that the “Closeness” between $\theta _L (S^i)$ and $\theta _L (S)$  should not be treated as an assumption for proving this Proposition, but an intermediate conclusion drawn from the proof. Please refer to Eq.(A.5) in the proof of Lemma A.1. This inequation shows that the closeness between $\theta _L (S^i)$ and $\theta _L (S)$  is automatically bounded by other arguments like Lipschitzness $\eta$ and dataset size $n$. Therefore, there’s no need to make extra definitions, and we have removed the “Closeness” assumption from the statement of Proposition 2.2.
>
> >**W2.2/Q2: The discussion about how large $\eta$ is in practice is missing**
>
> Thank you for your suggestion.
> The Lipschitz continuity depicts that the model function $L(f(\theta))$ has a good continuity in the parameter space. For general neural networks including LLMs, we typically assume that the output (loss) space of the model is relatively smooth, which means that large variations in the output are unlikely to occur within the neighborhood around any particular set of parameters. Especially during fine-tuning, the parameters being optimized are initialized at a relatively stable point (the optimum of the pre-training task is often located at a geometrically flatter region), thus it is proper to assume a fine continuity in the parameter space.
>
> Furthermore, we also conducted experiments to quantitatively present some empirical estimations on the value of \eta. We fine-tuned a LLaMA2-7B model and monitored the changes in parameters ($||\Delta \theta||$) and corresponding changes of losses (i.e., $\Delta L$) during the fine-tuning process. (as the direction of gradient descent indicates the largest loss variations at the current parameters, which is closer to the lower bound estimation of $\eta$)
> Based on the observed results, we could empirically estimate that the $\Delta L$ to $||\Delta \theta||$ ratio for a LlaMA2-7B model ranges from 0.01 to around 0.5. In other words, the lower bound of $\eta$ is around 0.5 empirically, which is a relatively small constant.
>
> >**W3: Mismatch between the assumption in Proposition 3.1 and its proof**
>
> We are sorry for this mistake. In fact, the assumption in the proof in Appendix A.2 should be the same as the assumption in Proposition 3.1, i.e., the loss function is convex w.r.t . the final activation of the model. Such an assumption is realistic since most loss functions used for training, like cross-entropy and MSE, are convex w.r.t to their inputs. We have fixed this mistake in the proof from Appendix A.2 in our revised paper. Thank you again for pointing it out.
>
> >**W4: Mistakes made in writing**
>
> Thank you for your suggestions and corrections. We followed your advice and improved our writing in the revised paper.

---

> > ### Comment · Reviewer_jLmX · 2024-11-26
> > **About the authors' response**
> >
> > Thank you for your further clarifications. I've also read the revised version of the submision. The writing seems being improved. However, many unclear notations or definitions remain missing. More importantly,
> >
> > 1. I am not convinced about their explanation about the role of "closeness" in their analysis. I read one more time on their revised version, and then detected one more thing that seems to be wrong or inconclusive. This point relates to my question about "closeness" before.
> >
> > In their proof, from line 728, a Taylor approximation is used to **approximate** the loss at a point $\theta_L(S)$. Then such an approximation is used to derive the stability constant. Now the "closeness" plays an important role here, **since Taylor approximation may be bad at far points and only good at very close points.** The authors completely ignore the approximation error, and use the approximation as if the error is 0. This is unsound. It is well-known that the error of the second-order Taylor approximation is $O(\\| \theta_L(S) - \theta_L(S^i) \\|^2)$.
> >
> > 2. The new proof of Proposition 3.1 is still wrong. The authors use Jensen inequality to move the loss from the outside to the inside positions of an expectation. Such a movement is possible for convex losses. However it is well-known that the training loss for DNNs is often nonconvex, and hence their movement is not well-supported. The key mistake can come from the use of the general notation $\theta$ to represent a model.
> >
> > I think the authors need a significant effort to improve their theoretical analysis and writing. From those reasons, I keep my current score.

---

> ### Author Response · Authors · 2024-11-28
> **Response to Reviewer jLmX #2**
>
> Thank you for your time and efforts in considering our rebuttal. We have made further adjustments to our manuscript according to your suggestions and presented detailed explanations point-by-point as below.
>
> >**Questions about “closeness”:**
>
> First of all, we’d like to genuinely apologize for our mistakes in the previous modification. The assumption of closeness between $\theta_L(S^i) $ and $\theta_L(S) $ is indeed necessary as the error of second-order Taylor expansion could not be ignored if the closeness is not ensured. Hence, in this updated version, we recover the original assumption with a more specific constraint, i.e., $||\theta_L(S^i) - \theta_L(S)|| \leq \epsilon \rightarrow 0$. Under this assumption, the correctness of Lemma 1 is ensured as the error of Taylor expansion in Eq.(A.1) is $O(\epsilon^3)$ and can be eliminated for a small $\epsilon$. here we make further clarifications on: 1) the practical reasonableness of the closeness assumption, 2) how the theoretical results of PHS stability (Lemma 1) connect with the closeness assumption.
>
> 1) **Practical reasonableness of the closeness assumption**: In this paper, we are dealing with the fine-tuning scenario. Therefore, the algorithms optimizing $\theta_L(S^i) $ and $\theta_L(S) $ are both initialized with the same set of parameters $\theta_0 $ (or zero-initialization of LoRA components $\Delta \theta$). Moreover, in LoRA fine-tuning, we also take a relatively small learning rate (e.g., 5e-5 on LLaMA2-7B), and the datasets only differ in one sample. All the factors above determine that the optima, $\theta(S^i) $ and $\theta(S)$, will be very close empirically. In conclusion, the way we translate the “closeness” assumption into practice is to select a small learning rate and a zero-initialization of $\Delta \theta$.
>
> 2) **How the theoretical results of PHS stability (Lemma 1) connect with the closeness assumption**: As shown in Eq.(A.5), we obtain another upper bound of the “closeness” except for $\epsilon$, and is further derived to obtain the final result of Lemma 1. Here, we clarify that, with a sufficiently large $\lambda$ (i.e., $\lambda \geq \frac{\eta}{n\epsilon} - \frac{1}{2}\Lambda_{\min}$), this bound is tighter than $\epsilon$. This means that under the assumption of the closeness of $||\theta_L(S^i) - \theta_L(S)|| $ with a constant $\epsilon$, the optima could be closer than originally assumed if the regularization strength is sufficiently large. Meanwhile, this result also ensures the tightness of our final result in Lemma 1, i.e., the PHS stability is no larger than $\eta\epsilon$ (derived by plugging $\lambda$ conditions into (A.7)). The results also guide us to ensure the $\lambda$ is sufficiently large in practice, where we are actually applying **hard** dropout mechanisms and this could be regarded as a very large regularization strength.
>
> In conclusion, the current version of our theoretical results ensures: a) correctness, b) tightness, c) practical reasonableness.
>
> >**Questions about Proposition 3.1:**
>
> Here we’d like to clarify that we **only assume the loss function (cross-entropy, MSE, …) is convex w.r.t. the model output $o$**. This is a very natural assumption as it is true of mainstream loss functions. We also make our proofs more detailed in our updated version, where we proved for any distribution $O$ of outputs $o$, the Jensen inequality holds, then for any distribution $\mathcal{O}(x) = M(x; \mathcal{D}$), taking expectations of $x$ and $y$, the inequality still holds. We have further adjusted our proofs with more details in Appendix A.2.
>
> Thank you once again for your careful reading and profound suggestions. We believe the clarity and quality of our theoretical results have been promoted in the new version of our manuscript.

---

> ### Author Response · Authors · 2024-12-01
> **A kind reminder as the deadline is approaching**
>
> Dear Reviewer jLmX,
>
> I hope this message finds you well.
>
> I am writing to kindly remind you that the deadline for the discussion phase regarding our paper is drawing close. We greatly appreciate the time and effort you have already dedicated to reviewing our work. Your insights and feedback are invaluable to us, and we believe they have significantly enhanced the quality of our manuscript.
>
> We are wondering if we have addressed your concerns in our previous responses, and we are more than willing to provide any additional information or clarification as assistance.
>
> Thank you once again for your time and consideration. We are looking forward to hearing from you soon.
>
> Best Regards.

---

> > ### Comment · Reviewer_jLmX · 2024-12-03
> > **Response about the new revision**
> >
> > Thank you for your further clarifications. I've also read the revised version of the submision. The writing seems being improved further. The writing and proof of Proposition 3.1 seems better now.
> >
> > However, the presentation about the theoretical part in Section 2.2 is not very accurate. Although there's some modification about Proposition 2.2, its proof is mostly the same as the previous version. That is, the error of Taylor approximation is completely ignored, and the autors use the approximation as if the error is 0. Some of my concerns about the writing in Section 2.2 was not improved, including the notations, definitions. More importantly, the mixed uses of the parameters for model and learning algorithm in their writing pose difficulties for accurate understanding. This also rises the accuracy of their analysis.
> >
> > I suggest to remove the theoretical part for future revision. For the current version, I am not convinced enough to change the score.

---

### Official Review · Reviewer_Re4B · 2024-11-03

**Soundness:** 3
**Presentation:** 3
**Contribution:** 3
**Rating:** 6
**Confidence:** 4

**Summary:**

The paper introduces LoRA Dropout, a refined dropout strategy applied to the rows and columns of LoRA's low-rank matrices, which enhances sparsity patterns without significant computational overhead. Additionally, the authors propose a test-time ensemble strategy that aggregates outputs from different dropout instances, further improving generalization. Experiments on NLP tasks like GLUE and SQuAD benchmarks demonstrate that LoRA Dropout significantly narrows the training-test loss gap and enhances model performance and generalization.

**Strengths:**

1. The paper is well-executed, with thorough theoretical analysis supporting the proposed method.
2. The paper is generally clear and well-structured, making it accessible to readers familiar with PEFT and regularization techniques.

**Weaknesses:**

1. Theoretical Assumptions and Generalizability
The theoretical framework relies on specific assumptions such as η-Lipschitz continuity and positive semi-definiteness of the Hessian matrix. These conditions may not universally apply to all language models or downstream tasks, making the theoretical guarantees less generalizable
2. Computational Efficiency and Practicality
The paper acknowledges that sampling multiple dropout instances during training and testing increases computational costs but lacks quantitative evidence on the extent of this overhead. Without this data, it is difficult for practitioners to evaluate the feasibility of adopting LoRA Dropout in real-world applications.
3. Experimental diversity
Although relevant experiments have been conducted in Lora and AdaLora, their effectiveness in other Lora variants has not been demonstrated.

**Questions:**

1. Can the authors provide more clarity on the assumptions made for the theoretical analysis, specifically the \eta-Lipschitz continuity and the positive semi-definiteness of the Hessian? Are there common NLP models or settings where these conditions do not hold?
2. Could the authors provide more detailed quantitative data on the computational overhead introduced by sampling multiple dropout instances during training and testing? How significant is the increase in training and inference time compared to the original LoRA or other baselines?
3. Can the author provide more experiments on other variants of Lora to demonstrate the effectiveness of the proposed Loradropout framework? E.g., LoRA+, VeRA, LoRA- fa.
4. What is the difference between the author's Lora dropout method and other existing dropout methods, such as DropKey, DropAttention, and HiddenCut.

---

> ### Author Response · Authors · 2024-11-26
> **Rebuttal to Reviewer Re4B (1)**
>
> Thanks for all the constructive suggestions and comments concerning our paper.  We made every effort to address all the concerns and have uploaded a revised version of the paper. In the following, we quote your comments and then give our detailed response point-by-point.
>
> > **W1/Q1:Detailed Explanations on Lipschitz continuity condition on parameters and positive semi-definiteness of Hessian**
>
>  **Lipschitz continuity**: This condition depicts that the model function $L(f(\theta))$ has a good continuity in the parameter space. For general neural networks including LLMs, We typically assume that the output (loss) space of the model is relatively smooth, which means that large variations in the output are unlikely to occur within the neighborhood around any particular set of parameters. Especially during fine-tuning, the parameters being optimized are initialized at a relatively stable point (the optimum of the pre-training task is often located at a geometrically flatter region), thus it is proper to assume a fine continuity in the parameter space.
>
> We also quantitatively present some empirical estimations on $\eta$. We fine-tuned a LLaMA2-7B model and monitored the changes in parameters ($||\Delta \theta||$) and corresponding changes of losses (i.e., $\Delta L$) during the fine-tuning process. (as the direction of gradient descent indicates the largest loss variations at the current parameters, which is closer to the lower bound estimation of $\eta$)
> Based on the observed results, we could empirically estimate that the $\Delta L$ to $||\Delta \theta||$ ratio for a LlaMA2-7B model ranges from 0.01 to around 0.5. In other words, the lower bound of $\eta$ is around 0.5 empirically, which is a relatively small constant.
>
> **Positive semi-definiteness of Hessian**: We admit that this assumption is too strong. Actually, we do not require such a strong assumption to support our proof. In detail, we only require $\Lambda + 2\lambda$ to be non-negative in the proof procedure (A.1) ($\Lambda$ denotes the eigenvalues of Hessian through unitary decomposition $Udiag{\Lambda}U^{-1}$). This means that we only need to constrain the regularization coefficient $\lambda$ to be sufficiently large (i.e., $\lambda \geq -\frac{1}{2}\Lambda_{\min}$), instead of hypothesizing an overly strong property of the optimized function. In fact, in our implementation, the regularization is accomplished in a “harder” way through dropout instead of the sparsity term, which complies with the setting of a large regularization strength $\lambda$. In the new version of our manuscript, we have modified the corresponding condition to the constraints on regularization strength. Thank you again for your suggestions.
>
> > **W2/Q2: Details on the computational overhead and practicality**
>
> We have reported the quantitative stats of training time on GLUE in Table C.2 (Appendix C). The sampling of multiple dropout instances will indeed increase the training and testing time, and the increase is related to the number of samples.
>
> However, the computational overhead would not make LoRA Dropout unusable in practice. Practically, the overfitting problem will become severe only when the data amount is limited. Under this scenario, the training time will not be a great problem (training time is short for a limited data scale) in comparison with an unacceptable overfitted performance. This also means one can adjust the number of samples to conduct a trade-off between overfitting control and training/inference time cost on their demand.
>
> >**W3/Q3: More experiments with other LoRA variants**
>
> Thanks for the suggestions. Here, we provide the results on more LoRA variant models.
>
> **Results on LoRA+:**
>
> |                         | MNLI  | SST2  | COLA  | QQP   | QNLI  | RTE   | MRPC  | STSB  | Avg   |
> |-------------------------|-------|-------|-------|-------|-------|-------|-------|-------|-------|
> | LoRA+                   | 90.48 | 94.03 | 68.93 | 91.41 | 94.00 | 86.28 | 88.97 | 91.36 | 88.18 |
> | LoRA+ w/ LoRA Dropout   | **90.64** | **95.76** | **71.02** | **91.76** | **94.63** | **88.45** | **91.67** | **92.04** | **89.50** |
>
> **Results on LoRA-FA:**
>
> |                         | MNLI  | SST2  | COLA  | QQP   | QNLI  | RTE   | MRPC  | STSB   | Avg   |
> |-------------------------|-------|-------|-------|-------|-------|-------|-------|--------|-------|
> | LoRA-FA                 | 89.84 | 94.04 | 70.01 | 91.52 | 93.78 | 86.64 | 90.20  | 91.58  | 88.45 |
> | LoRA-FA w/ LoRA Dropout | **90.78** | **95.87** | **71.31** | **91.81** | **94.76** | **89.17** | **90.93** | **91.91**  | **89.57** |
>
> The results demonstrate the effectiveness of LoRA Dropout in different variants of LoRA.

---

> ### Author Response · Authors · 2024-11-26
> **Rebuttal to Reviewer Re4B (2)**
>
> >**Q4. Differences between LoRA Dropout and other dropout methods**
>
> The differences mainly lie in two aspects:
> 1. Model architecture. Other dropout methods, like DropKey, DropAttention, and HiddenCut, are designed specifically for the transformer architecture. LoRA Dropout is designed for the PEFT method LoRA, with no restriction to the model architecture.
> 2. Perturbation space. Dropout methods, like DropKey, DropAttention, and HiddenCut, conduct dropout **in the input space**, e.g., dropping random elements (tokens) from the sample’s self-attention matrix. Therefore, these dropout methods increase the diversity of samples and could be viewed as a kind of data augmentation method. LoRA Dropout conducts dropout **in the parameter space**. With theoretical support, LoRA dropout prevents overfitting by introducing proper sparsity regularization during training. In conclusion, the perturbation spaces of LoRA dropout and other mentioned methods are actually orthogonal. We regard studying the effect of combining both techniques as an interesting future work.

---

### Note · Authors · 2025-01-20

I have read and agree with the venue's withdrawal policy on behalf of myself and my co-authors.